# Revisiting Coding-Based Approaches to Overcome the Curse of Dimensionality in Learning-Based Watermarking

Yupeng Qiu [1]  Han Fang [2]  Ee-Chien Chang [1]

## Abstract

Deep learning–based watermarking has substantially improved robustness to real-world noise, but its performance degrades as the payload dimension increases. In contrast, coding-based methods such as quantization index modulation (QIM) do not suffer from this curse of dimensionality, although they are less robust to real-world noise. To leverage the strengths of both approaches, we propose OrthoMark, a framework that decouples robust feature extraction from message encoding. OrthoMark first learns a distortion-invariant feature representation using a deep robust feature extractor, and then performs watermark encoding and decoding in this feature domain using coding-based methods. Extensive experiments demonstrate that OrthoMark significantly improves the trade-off among visual quality, robustness, and capacity compared to prior deep watermarking methods, with particularly large gains in the high capacity regime, effectively overcoming the curse of dimensionality. Our code is available at https://github.com/QQiuyp/OrthoMark.

## 1. Introduction

Image watermarking is widely used for copyright protection (Van Schyndel et al., 1994; Cox et al., 2001) and provenance tracking (Fernandez et al., 2023; Wen et al., 2023; Yang et al., 2024). In practice, a watermarking method should strike an effective balance among three competing objectives: *high visual quality*, meaning the watermarked image remains perceptually indistinguishable from the host image; *strong robustness*, ensuring the encoded information

[1]School of Computing, National University of Singapore, Singapore [2]School of Cyber Science and Technology, University of Science and Technology of China, Hefei, Anhui, China. Correspondence to: Han Fang <fanghan@ustc.edu.cn>.

*Proceedings of the 43rd International Conference on Machine Learning*, Seoul, South Korea. PMLR 306, 2026. Copyright 2026 by the author(s).

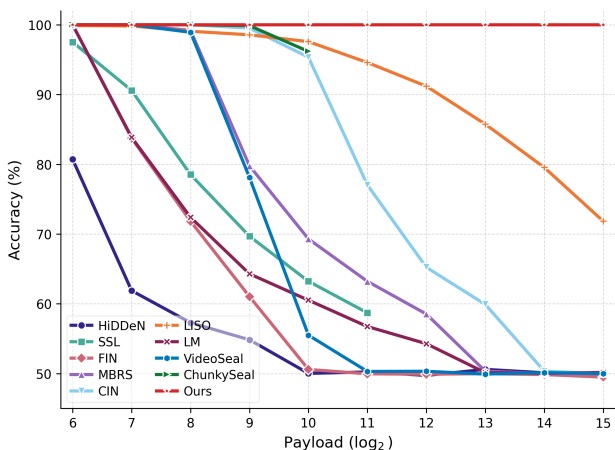

*Figure 1.* **Decoding accuracy under increasing payload. (PSNR is set to 50 dB for all methods.)** Bit decoding accuracy (%) on clean (undistorted) watermarked images as the payload increases (reported as $\log_2$ of the payload dimension) for state-of-the-art deep watermarking methods and our method. Existing methods exhibit a pronounced accuracy collapse in the high capacity regime, whereas our method sustains near-perfect decoding accuracy regardless of the payload dimension.

survives after the watermarked image undergoes distortions; and *high capacity*, allowing more information to be reliably encoded and decoded.

Classical coding-based watermarking methods typically rely on hand-crafted encoding and decoding rules, such as quantize-and-replace schemes (e.g., LSB) (Van Schyndel et al., 1994), additive spread spectrum (Cox et al., 1997), and quantization index modulation (QIM) (Chen & Wornell, 2002). Among these, QIM is particularly effective. Under additive white Gaussian noise (AWGN), its distortion-compensated variant (DC-QIM) is capacity-achieving, meaning that it attains the information-theoretic maximum achievable capacity (Chen & Wornell, 2002). Although coding-based watermarking methods perform well under additive white Gaussian noise (AWGN), their performance degrades significantly in real-world environments, where noise deviates substantially from AWGN. One approach to address this issue is to apply watermarking in a transform domain (e.g., the wavelet domain), in which real-world noise can be more closely approximated as AWGN.

Despite notable progress, designing such accurate transformations remains challenging.

Recently, deep learning–based watermarking methods have demonstrated strong robustness against diverse and complex distortions. By training end-to-end with a variety of noise layers, these models bypass the need for sophisticated coding designs and often outperform coding-based watermarking methods under real-world noise (Zhu et al., 2018; Jia et al., 2021; Chen et al., 2022; Fernandez et al., 2022; Ma et al., 2022; Fang et al., 2023; Fernandez et al., 2024; Qiu et al., 2025; Petrov et al., 2025).

The superior performance of learning-based watermarking over coding-based methods can be attributed to their ability to more accurately model real-world noise. However, coding-based methods can achieve theoretical optimality under AWGN by efficiently packing watermark codewords according to mathematical principles. Given that QIM encoding and decoding are inherently non-differentiable, this raises doubts about whether purely learning-based approaches can achieve similarly effective packing. Supporting this concern, in a noiseless setting, coding-based methods can attain perfect decoding accuracy and thus are able to attain arbitrarily large capacity, limited only by floating-point precision. In contrast, such behavior is not observed in learning-based methods. Empirically, increasing the payload in learning-based methods often results in an abrupt collapse in decoding accuracy and eventually leads to a lower capacity, even when the watermarked content is free of noise, as shown in Fig. 1.

This leads to the main research question of this paper: *is it possible to combine the strengths of learning-based methods in modeling diverse and complex real-world noise with the intrinsic structure of coding-based methods to further enhance performance?*

We begin by identifying a key bottleneck in learning-based methods, which we argue stems from the curse of dimensionality in learning efficient packing over an exponentially large message space. As the number of possible messages grows exponentially with payload dimension, learning an efficient packing from a comparatively slow-growing number of training samples becomes increasingly difficult. This leads to increased coding cross-talk, where different message bits are conveyed by non-orthogonal, overlapping carriers. As the payload dimension continues to grow, the intensity of cross-talk rises, eventually causing a collapse in decoding accuracy.

One way to integrate the strengths of both approaches is to leverage machine learning to learn an accurate transformation from real-world noise to AWGN, so that coding-based methods can be applied. However, our preliminary attempts did not achieve the desired outcome, largely due to the difficulty of aligning the encoding and decoding processes. To overcome this challenge, we propose OrthoMark, a framework that combines coding-based and learning-based methods by decoupling feature-domain extraction from watermark encoding and decoding.

OrthoMark comprises two modules: (i) a deep Robust Feature Extractor (RFE) module that learns a distortion-invariant feature domain, and (ii) a Structured Encoding and Decoding (SED) module, inspired by QIM-style designs, that performs coding-based watermarking in this learned feature domain. SED leverages quantization-coset structures to support flexible encoding strength, improving visual quality. Meanwhile, its orthogonal bit-carrier construction suppresses coding cross-talk and enables scalable capacity. Overall, OrthoMark reconciles high visual quality, strong robustness, and high capacity within a unified framework.

The main contributions of this paper are as follows:

1. We uncover a fundamental limitation of learning-based watermarking: decoding performance degrades as the payload dimension increases. We attribute this failure to a curse of dimensionality that leads to coding cross-talk.
2. We propose OrthoMark, which combines the strengths of learning-based methods in modeling diverse and complex real-world noise with the intrinsic structure of coding-based methods.
3. Extensive experiments show that OrthoMark consistently improves the joint trade-off among visual quality, robustness, and capacity over prior deep watermarking methods, with particularly strong gains in the high capacity regime.

## 2. Related Work

### 2.1. Coding-based watermarking

Classical coding-based watermarking methods rely on hand-crafted encoding and decoding mechanisms. Early pixel-domain methods (Van Schyndel et al., 1994; Nikolaidis & Pitas, 1998) commonly used quantize-and-replace strategies, most notably least significant bit (LSB) modulation (Van Schyndel et al., 1994), which can achieve high capacity but is vulnerable to distortions. Additive spread spectrum (Cox et al., 1997) encodes messages by adding a low-energy pseudo-noise pattern and decodes via correlation. To obtain more controllable trade-offs between visual quality, capacity, and robustness, quantization-based encoding mechanisms were developed, most notably quantization index modulation (QIM) (Chen & Wornell, 2002). QIM exploits the host image as side information available at the encoder and encodes bits by selecting among quantization structures, resulting in enhanced overall performance. Moreover, QIM explicitly enforces independent bit carriers through a

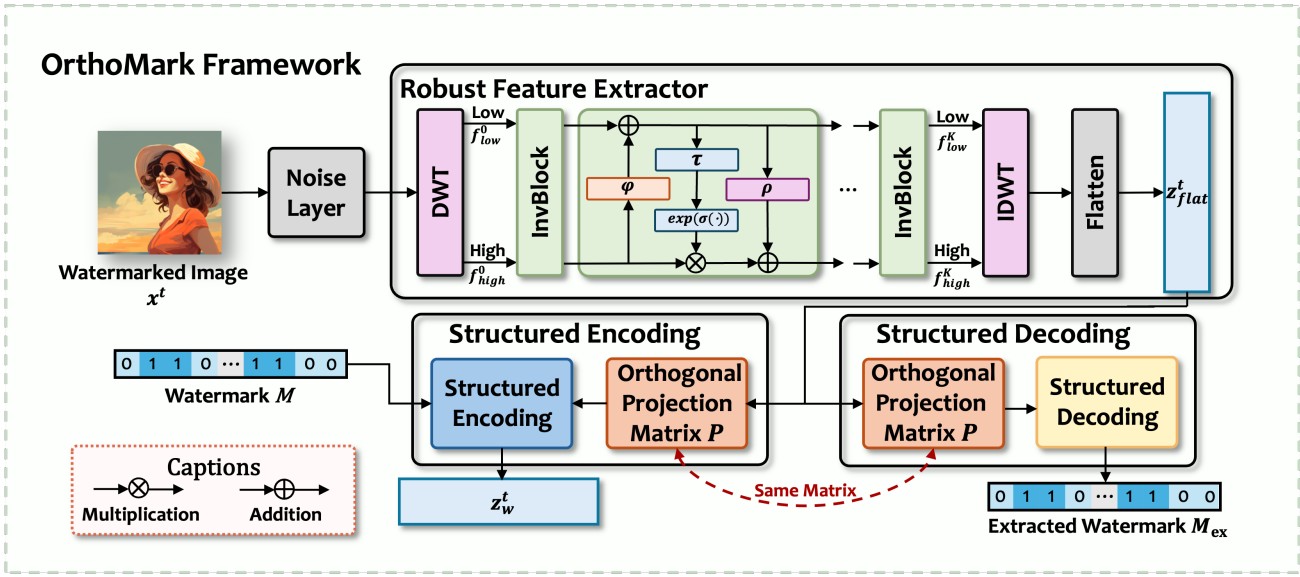

*Figure 2.* **Overview of the proposed OrthoMark framework.** OrthoMark decouples robust feature extraction from watermark encoding and decoding. Given a host image, a noise layer simulates distortions, and the Robust Feature Extractor (RFE; implemented as an invertible network with DWT/IDWT blocks) produces a distortion-invariant robust feature representation. The Structured Encoding and Decoding (SED) then performs watermark coding in this feature space: the watermark is encoded by structured encoding with an orthogonal projection matrix $P$ and decoded using the same $P$ to avoid encoder–decoder mismatch. The final watermarked image is obtained by optimizing the host image to match the target watermarked features over $T$ optimization iterations.

structured, highly non-linear quantization-coset geometry, which suppresses coding cross-talk; under the additive white Gaussian noise (AWGN), its distortion-compensated variant (DC-QIM) is capacity-achieving.

However, these pixel-domain classical watermarking methods are highly vulnerable to real-world noise. To improve robustness, classical methods instead encode watermarks in transform domain, such as the discrete cosine transform (DCT) (Bors & Pitas, 1996; Barni et al., 1998; Ahmidi & Safabakhsh, 2004), discrete wavelet transform (DWT) (Xia et al., 1998; Barni et al., 2001; Daren et al., 2001), or discrete Fourier transform (DFT) (Urvoy et al., 2014; Larbi et al., 2018), sometimes combined into hybrids (Hamidi et al., 2018). While such hand-crafted domains can perform well under the specific noise they are designed for, constructing and tuning them typically requires substantial expert effort, and it remains difficult to make them robust to the diverse and complex noise encountered in real-world settings.

### 2.2. Deep learning-based watermarking

Deep learning-based watermarking has advanced rapidly in recent years, achieving stronger robustness against diverse and complex noise. For better robustness, Zhu et al. (2018) introduced the Encoder–NoiseLayer–Decoder (END) framework, inserting differentiable noise layers between the encoder and decoder so robustness can be learned end-to-end. Building on this idea, Jia et al. (2021) proposed

MBRS, which mixes simulated and real JPEG compression to better handle JPEG compression. To address non-differentiable distortions, Zhang et al. (2021) introduced an attack-simulation layer (ASL) that approximates such effects while preserving gradient flow. Beyond END-style pipelines, Fang et al. (2023) and Ma et al. (2022) leveraged invertible neural networks (INNs) to better couple encoding and decoding, and Ma et al. (2024) designed a hybrid backbone combining Swin Transformers with deformable convolutions to strengthen geometric robustness. Qiu et al. (2025) further improved robustness in lightweight models via a decoding-oriented surrogate loss, and Fernandez et al. (2024) set a stronger efficiency–robustness baseline.

Despite these advances, capacity remains a major bottleneck for deep watermarking. While Chen et al. (2022) increases payload, it comes with substantial sacrifices in visual quality and robustness, limiting practicality. More broadly, most deep methods encode only a few hundred bits per image, and decoding accuracy often collapses as the payload increases, even in distortion-free settings, as shown in Fig. 1. This suggests that the bottleneck stems from the model itself rather than from external distortions. How to combine the complementary strengths of coding-based watermarking and deep watermarking remains an important challenge.

## 3. Problem Formulation

### 3.1. Linear superposition model

To analyze interactions among bits, we introduce a simple linear superposition model. For a fixed host image $I_{\text{ho}} \in \mathbb{R}^m$, we model encoding each bit as adding a perturbation along a corresponding carrier direction.

**Assumption 3.1** (Linear superposition model). For a given host image $I_{\text{ho}}$ and message $M$, the encoder $E$ outputs a watermarked image $I_{\mathbf{w}}$ of the form

$$I_{\mathbf{w}} = I_{\text{ho}} + \sum_{k=1}^{n} d_k(I_{\text{ho}}) \, M_k \, \mathbf{u}_k(I_{\text{ho}}), \qquad (1)$$

where $M_k$ denotes the $k$-th bit of $M$. $\mathbf{u}_k(I_{\text{ho}}) \in \mathbb{R}^m$ denotes the (unit-norm) carrier associated with bit $k$ for the host image $I_{\text{ho}}$, and $d_k(I_{\text{ho}}) > 0$ is the corresponding encoding strength. The message bits satisfy $M_k \in \{-1, +1\}$. For notational simplicity, we write $d_k$ and $\mathbf{u}_k$ as shorthand for $d_k(I_{\text{ho}})$ and $\mathbf{u}_k(I_{\text{ho}})$. In this analysis, we focus on high capacity decoding accuracy and temporarily set aside visual quality considerations. Accordingly, to simplify the analysis, we upper-bound the per-bit strengths by their maximum and replace $d_k$ with $d_{\max} \triangleq \max_{k \in \{1,\ldots,n\}} d_k$.

$$I_{\mathbf{w}} = I_{\text{ho}} + d_{\max} \sum_{k=1}^{n} M_k \, \mathbf{u}_k \qquad (2)$$

### 3.2. Coding cross-talk and carrier mismatch

Recall that the decoder maps a (possibly noised) input image to real-valued scores, which are then quantized into bits. Here we focus on the ideal (noiseless) setting and consider a decoder whose $i$-th score is obtained by projecting the watermarked image $I_{\mathbf{w}}$ onto a decoder-side carrier $\mathbf{u}_i'$:

$$y_i' = \langle I_{\mathbf{w}}, \mathbf{u}_i' \rangle. \qquad (3)$$

Here $\langle \cdot, \cdot \rangle$ denotes the inner product. Under the linear superposition model in (2), we have

$$y_i' = \langle I_{\text{ho}}, \mathbf{u}_i' \rangle + d_{\max} \sum_{k=1}^{n} M_k \langle \mathbf{u}_k, \mathbf{u}_i' \rangle. \qquad (4)$$

Let $\tilde{x}_i' \triangleq \langle I_{\text{ho}}, \mathbf{u}_i' \rangle$ and $\alpha_{ki} \triangleq \langle \mathbf{u}_k, \mathbf{u}_i' \rangle$. Equivalently,

$$y_i' = \tilde{x}_i' + d_{\max} M_i \alpha_{ii} + d_{\max} \sum_{k=1, \, k \neq i}^{n} M_k \alpha_{ki}, \quad (5)$$

For $i$-th bit, we decompose the score into a useful term and an interference term:

$$y_i' = \hat{d}_i \, M_i \, \alpha_{ii} + \Gamma_i, \qquad \Gamma_i \triangleq d_{\max} \sum_{k=1, \, k \neq i}^{n} M_k \alpha_{ki}, \quad (6)$$

where $\hat{d}_i \triangleq d_{\max} + \frac{M_i \tilde{x}_i'}{\alpha_{ii}}$. This decomposition assumes $\alpha_{ii} \neq 0$, which is mild for decoding: $\alpha_{ii}$ measures the alignment between the encoder carrier for bit $i$ and the corresponding decoder-side carrier, and $\alpha_{ii} = 0$ would mean that the useful signal for bit $i$ is completely invisible to the decoder. The term $\Gamma_i$ captures the aggregate bit-wise interference arising from (i) non-orthogonal carrier correlations across bits and (ii) possible mismatch between the encoder and decoder carriers.

**Optimistic interference setting.** Given i.i.d. symmetric message bits $\{M_k\}_{k=1}^{n}$ taking values in $\{-1, +1\}$, define

$$\alpha_{\min}^i \triangleq \min_{k \in \{1,\ldots,n\} \setminus \{i\}} |\alpha_{ki}|. \qquad (7)$$

In the most optimistic regime, we assume that all cross-bit correlations with bit $i$ attain this minimal magnitude, i.e., $|\alpha_{ki}| = \alpha_{\min}^i$ for all $k \neq i$. The resulting optimistic interference satisfies

$$\Gamma_i^{\text{opt}} = d_{\max} \alpha_{\min}^i \sum_{k=1, \, k \neq i}^{n} M_k. \qquad (8)$$

Moreover, by the central limit theorem (CLT), for sufficiently large $n$, $\Gamma_i$ can be well approximated by a zero-mean Gaussian random variable; details are deferred to Appendix A.1.

$$\Gamma_i^{\text{opt}} \sim \mathcal{N}\left(0, \, d_{\max}^2 \left(\alpha_{\min}^i\right)^2 (n-1)\right). \qquad (9)$$

Since the cross-bit interference is captured by $\text{Var}(\Gamma_i^{\text{opt}})$, it is natural to quantify, for each bit $i$, how detectable its contribution is in the presence of all other encoded bits. This leads to a per-bit signal-to-interference ratio (SIR), which compares the effective signal of bit $i$ against the aggregate multi-bit interference.

**Definition 3.2** (Signal-to-interference ratio). For bit $i$, the useful contribution in (6) is $\hat{d}_i M_i \alpha_{ii}$, while the optimistic internal interference is modeled by $\Gamma_i^{\text{opt}}$. We define the per-bit signal-to-interference ratio

$$\text{SIR}_i \triangleq \frac{\left(\hat{d}_i M_i \alpha_{ii}\right)^2}{\text{Var}\left(\Gamma_i^{\text{opt}}\right)} = \frac{\hat{d}_i^2 \alpha_{ii}^2}{d_{\max}^2 \left(\alpha_{\min}^i\right)^2 (n-1)}. \quad (10)$$

**Proposition 3.3** (Bit error probability vs. SIR). *Under the optimistic interference setting, the decoding score for bit $i$ admits the decomposition $y_i' = \hat{d}_i M_i \alpha_{ii} + \Gamma_i^{\text{opt}}$, where $\Gamma_i^{\text{opt}}$ is approximately zero-mean Gaussian with variance $\text{Var}(\Gamma_i^{\text{opt}})$ by the central limit theorem. The decoder recovers the bit via $\hat{M}_i = \text{sign}(y_i)$. Consequently, the bit error probability satisfies*

$$P_{e,i} \triangleq \mathbb{P}(\hat{M}_i \neq M_i) = Q\left(\sqrt{\text{SIR}_i}\right), \qquad (11)$$

*where $Q(\cdot)$ is the standard Gaussian Q-function, $Q(x) = \frac{1}{\sqrt{2\pi}} \int_x^\infty e^{-u^2/2} \, \mathrm{d}u$. In particular, $\mathrm{SIR}_i \to 0$ implies $P_{e,i} \to \frac{1}{2}$. See Appendix A.2.*

This result shows that, even without external distortions and setting aside visual quality, decoding accuracy is governed by the correlations between the encoder and decoder carriers. Under a fixed encoding strength, reliable high capacity decoding (large $n$) requires keeping $\mathrm{SIR}_i$ large. Concretely, this calls for a structured design that (i) maximizes the useful alignment $\alpha_{ii}$ and (ii) minimizes aggregate bit-wise interference by reducing cross-correlations $|\alpha_{ki}|$ for $k \neq i$ so that $\mathrm{Var}(\Gamma_i)$ remains small, which is a requirement that deep watermarking methods typically struggle to learn.

## 4. Methodology

As analyzed above, the capacity of deep watermarking is constrained by two structural limitations. **(i) Encoder–decoder misalignment**: the encoder and decoder typically differ in architectures, and are trained jointly under asymmetric objectives, so the useful projection $\alpha_{ii}$ cannot be explicitly controlled. **(ii) Non-orthogonal multi-bit encoding**: per-bit carriers $\{\mathbf{u}_k\}_{k=1}^n$ are learned jointly through shared representations, and standard losses do not regularize the cross-correlations $|\alpha_{ki}|$ ($k \neq i$), which accumulate into multi-bit interference as payload grows. In contrast, classical coding-based watermarking (Van Schyndel et al., 1994; Cox et al., 1997; Chen & Wornell, 2002) resolves both issues by explicit design: matched decoders maximize $\alpha_{ii}$, and orthogonal carriers suppress $|\alpha_{ki}|$ (see Appendix A.3). However, its robustness is limited by the difficulty of hand-crafting feature domains that remain stable under complex distortions. These complementary strengths motivate OrthoMark: we use a deep network to learn a distortion-invariant feature domain, and perform watermark encoding and decoding in this domain via structured, coding-based mechanisms.

### 4.1. OrthoMark Framework

**Overview.** OrthoMark consists of two modules: a Robust Feature Extractor (RFE) that maps an image to a distortion-invariant feature, and a Structured Encoding and Decoding (SED) module that performs message coding in this feature domain. Together, they support three operational stages. **Encoding** maps a host image $I_{\mathrm{ho}}$ and a binary message $M$ to a target watermarked feature $z_{\mathrm{w}}$: the RFE first extracts a robust feature $z$, and SED then projects $z$ onto an $n$-dimensional subspace via an orthogonal projection matrix $P$ and quantizes each projected coordinate using QIM, with the quantizer indexed by $M$. **Decoding** recovers the message from a (possibly distorted) image: the same RFE extracts its robust feature, and SED reuses the same

$P$ to project the feature and assign each coordinate to its nearest quantizer, yielding $\hat{M}$. **Optimization** bridges the feature-domain target and the image domain: starting from $x^0 = I_{\mathrm{ho}}$, the image is iteratively updated for $T$ steps so that its robust feature aligns with $z_{\mathrm{w}}$. During *training*, the RFE parameters and the image are jointly optimized using three loss terms: visual quality, clean feature alignment, and robust feature alignment after the noise layer; during *inference*, the RFE is frozen and only the image is optimized to obtain the final watermarked image $I_{\mathrm{w}} = x^T$.

**Robust Feature Extractor.** Reliable watermarking requires a feature representation that remains stable under distortions, so messages can be encoded and decoded consistently. Compared with the pixel domain, transform-domain representations are often more stable. In OrthoMark, we extract features in the wavelet domain. Given a host image $I_{\mathrm{ho}} \in \mathbb{R}^{3 \times H \times W}$, we first apply a discrete wavelet transform (DWT) to obtain a low-frequency component $f_{\mathrm{low}} \in \mathbb{R}^{3 \times \frac{H}{2} \times \frac{W}{2}}$ and high-frequency components $f_{\mathrm{high}} \in \mathbb{R}^{9 \times \frac{H}{2} \times \frac{W}{2}}$.

We adopt an invertible neural network (INN) as the robust feature extractor (RFE) due to its bijective architecture, which preserves information flow and yields stable representations (Dinh et al., 2014; 2016; Kingma & Dhariwal, 2018). We use the INN as a feature extractor and do not call the inverse path during inference. The RFE consists of $K$ identical sub-modules. In the $k$-th sub-module, the inputs are $\left(f_{\mathrm{low}}^k, f_{\mathrm{high}}^k\right)$ and the outputs are $\left(f_{\mathrm{low}}^{k+1}, f_{\mathrm{high}}^{k+1}\right)$, defined as

$$\begin{aligned}
f_{\mathrm{low}}^{k+1} &= f_{\mathrm{low}}^k + \varphi\left(f_{\mathrm{high}}^k\right), \\
f_{\mathrm{high}}^{k+1} &= f_{\mathrm{high}}^k \odot \exp\left(\sigma\left(\rho\left(f_{\mathrm{low}}^{k+1}\right)\right)\right) + \tau\left(f_{\mathrm{low}}^{k+1}\right),
\end{aligned} \tag{12}$$

where $\varphi(\cdot)$, $\rho(\cdot)$, and $\tau(\cdot)$ are instantiated as dense blocks (Huang et al., 2017), $\sigma(\cdot)$ is a sigmoid function scaled by a constant factor to serve as a clamp, and $\odot$ denotes element-wise multiplication. After the final block, we apply the inverse DWT to the outputs to obtain the robust feature $z \in \mathbb{R}^{3 \times H \times W}$.

**Structured Encoding.** Deep watermarking often suffers from encoder–decoder mismatch and non-orthogonal bit carriers. Both factors induce bit-wise interference and make capacity difficult to scale up. To address this issue, we introduce a structured encoding module consisting of an explicit orthogonal projection matrix and a coding-based message encoding module. Let $z$ denote the robust feature produced by the RFE. We first flatten it into a vector $z_{\mathrm{flat}} \in \mathbb{R}^m$, and then project it onto an $n$-dimensional subspace using an orthogonal projection matrix $P$, which enforces

**orthogonal bit carriers** by construction.

$$z_{\mathrm{proj}} = z_{\mathrm{flat}}P,$$
$$\text{where } P \in \mathbb{R}^{m \times n} \text{ satisfies } P^\top P = I_n, \tag{13}$$

where $I_n$ denotes the $n \times n$ identity matrix.

The message encoding mechanism follows the QIM method (Chen & Wornell, 2002). Given a binary message $M \in \{0,1\}^n$ and the projected feature $z_{\mathrm{proj}} \in \mathbb{R}^n$, we produce the watermarked feature $z_{\mathrm{w}} \in \mathbb{R}^n$ via element-wise message-indexed quantization. Concretely, for each coordinate $i \in \{1, \ldots, n\}$, where $[\cdot]$ denotes indexing,

$$z_{\mathrm{w}}[i] = \begin{cases} \Delta \left\lfloor \dfrac{z_{\mathrm{proj}}[i] - d_0}{\Delta} \right\rceil + d_0, & \text{if } M_i = 0, \\[2mm] \Delta \left\lfloor \dfrac{z_{\mathrm{proj}}[i] - d_1}{\Delta} \right\rceil + d_1, & \text{if } M_i = 1, \end{cases} \tag{14}$$

where $\Delta > 0$ is the quantization step size, $\lfloor \cdot \rceil$ denotes rounding to the nearest integer, and $\{d_0, d_1\}$ are two dithers that index the quantizers, with $d_0 = 0$ and $d_1 = \Delta/2$.

**Structured Decoding.** The structured decoding module reuses the same orthogonal projection matrix $P$ as the structured encoding module to compute $z_{\mathrm{proj}}$, thereby avoiding **encoder–decoder mismatch**. It then extracts the message by selecting the quantizer index whose reconstruction is closest to $z_{\mathrm{proj}}$. Concretely, for each coordinate $i \in \{1, \ldots, n\}$, we decode

$$M_{ex}[i] = \arg\min_{b \in \{0,1\}} |z_{\mathrm{proj}}[i] - q_b[i]|,$$
$$q_b[i] = \Delta \left\lfloor \frac{z_{\mathrm{proj}}[i] - d_b}{\Delta} \right\rceil + d_b, \tag{15}$$

where $d_0 = 0$ and $d_1 = \Delta/2$.

## 4.2. Loss Function

Unlike prior deep watermarking frameworks, which use an explicit encoder to generate the watermarked image in a single forward pass, OrthoMark treats the host image as an optimizable variable and jointly optimizes it with the RFE via backpropagation. Starting from the host image $x^0 = I_{\mathrm{ho}}$, we perform $T$ iterative updates and obtain the final watermarked image $x^T = I_{\mathrm{w}}$. At each iteration $t \in \{1, 2, \ldots, T\}$, we minimize a training loss composed of three terms.

**Visual quality loss.** To preserve visual quality, the watermarked image should remain close to the host image. We therefore define the visual quality loss as

$$\mathcal{L}_{\mathrm{VQ}}^t = \mathrm{MSE}(x^t, x^0). \tag{16}$$

**Feature alignment loss.** We first consider watermark encoding in the distortion-free setting, where the intermediate watermarked image $x^t$ is fed into the RFE directly without any noise layer. Given the robust feature $z^t$ extracted by the RFE, the structured encoding module encodes the message $M$ into $z^t$ to produce the target watermarked feature $z_{\mathrm{w}}^t$. To guide $x^t$ to carry the message during iterative optimization, we minimize the following feature-alignment loss:

$$\mathcal{L}_{\mathrm{FA}}^t = \mathrm{MSE}(z^t, z_{\mathrm{w}}^t). \tag{17}$$

**Robust feature alignment loss.** To obtain robustness to various distortions, at each iteration $t$ we also pass the intermediate watermarked image $x^t$ through a noise layer $\mathcal{N}(\cdot)$ before the RFE, yielding a noised robust feature $z_{\mathrm{no}}^t$. Because our message encoding mechanism follows the QIM method in (14), encoding the same message $M$ into the clean feature $z^t$ and the noised feature $z_{\mathrm{no}}^t$ can produce different watermarked targets (i.e., $z_{\mathrm{w}}^t \neq z_{\mathrm{w,no}}^t$ in general). This target inconsistency across distortions makes the optimization objective vary with the sampled noise layer and can destabilize training. To stabilize training while encouraging distortion invariance, we use the clean watermarked feature $z_{\mathrm{w}}^t$ as a shared target for both the clean feature $z^t$ and the noised feature $z_{\mathrm{no}}^t$. Specifically, we set the target for $z_{\mathrm{no}}^t$ to be $z_{\mathrm{w}}^t$ obtained by encoding $M$ into the clean feature $z^t$, and define the robust feature alignment loss as

$$\mathcal{L}_{\mathrm{RFA}}^t = \mathrm{MSE}(z_{\mathrm{no}}^t, z_{\mathrm{w}}^t). \tag{18}$$

**Total training loss.** The total loss at iteration $t$ is a weighted sum of the visual quality loss, the feature alignment loss, and the robust feature alignment loss:

$$\mathcal{L}_{\mathrm{total}}^t = \lambda_1 \, \mathcal{L}_{\mathrm{VQ}}^t + \lambda_2 (\mathcal{L}_{\mathrm{FA}}^t + \mathcal{L}_{\mathrm{RFA}}^t). \tag{19}$$

Here $\lambda_1$ and $\lambda_2$ balance visual quality and robustness during optimization.

**Iterative optimization of the watermarked image.** Over the $T$ optimization steps, we update the RFE parameters using standard backpropagation. Unlike typical training in deep watermarking frameworks, OrthoMark also treats the host image $I_{\mathrm{ho}} (x^0)$ as an optimizable variable and obtains the final watermarked image $I_{\mathrm{w}} (x^T)$ via iterative optimization. Concretely, at iteration $t$ we update

$$x^{t+1} = x^t - \eta \, \nabla_{x^t} \mathcal{L}_{\mathrm{total}}^t, \tag{20}$$

where $\eta$ is the step size.

**Inference stage.** At the inference stage, we freeze the RFE and optimize only the input image, starting from the host image $I_{\mathrm{ho}} (x^0)$ and a given message $M$. Specifically, we compute the loss in (19) and iteratively update $x^t$ using (20) for $T$ steps to obtain the final watermarked image $I_{\mathrm{w}} (x^T)$.

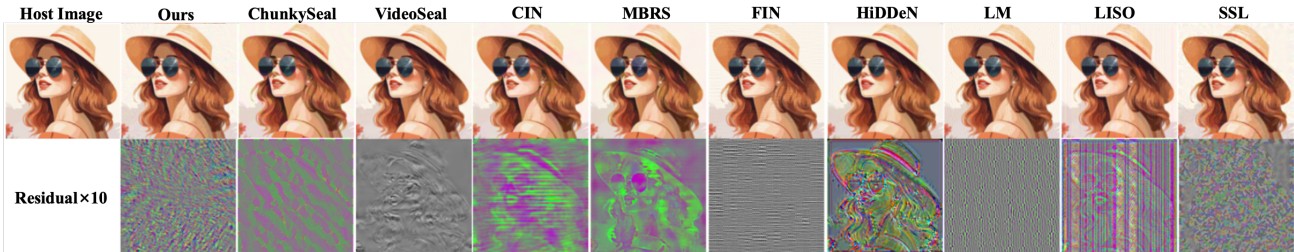

*Figure 3.* **Visual comparison of watermarked images.** Top: host images and watermarked images. Bottom: ×10 magnified residuals between watermarked and host images.

*Table 1.* Benchmark comparisons on visual quality and robustness against diverse distortions.

| Method | PSNR (dB) | Signal Distortions | | | | | | Geometric Transforms | | | | | Photometric Transforms | | | | AVG |
|---|---|---|---|---|---|---|---|---|---|---|---|---|---|---|---|---|---|
| | | JPEG | MF | GF | DP | S&P | GN | Erase | C&R | Shear | Rotate | Elastic | Hue | Bright | Contrast | Saturate | |
| DwtDctSvd | 35.49 | 65.99 | 83.30 | 87.72 | 85.79 | 69.73 | 86.82 | 70.39 | 50.85 | 51.29 | 50.02 | 56.62 | 47.05 | 53.09 | 52.80 | 52.92 | 64.29 |
| HiDDeN | 33.60 | 56.99 | 51.10 | 53.03 | 65.90 | 61.86 | 57.90 | 70.13 | 64.06 | 58.27 | 55.97 | 67.00 | 68.80 | 64.11 | 69.12 | 72.06 | 62.42 |
| LISO | 33.04 | 53.54 | 62.04 | 67.83 | 98.30 | 97.98 | 98.44 | 97.98 | 50.92 | 48.67 | 49.93 | 62.41 | 96.81 | 96.48 | 99.68 | 99.31 | 78.69 |
| SSL | 35.12 | 63.60 | 68.57 | 70.22 | 73.16 | 49.26 | 67.74 | 54.60 | 59.19 | 55.74 | 68.11 | 59.93 | 79.27 | 73.58 | 78.35 | 78.17 | 66.63 |
| FIN | 35.74 | 95.21 | 92.87 | 95.70 | 99.61 | 87.11 | 99.80 | 89.06 | 52.73 | 69.43 | 50.05 | 70.90 | 100 | 94.14 | 98.73 | 100 | 86.36 |
| MBRS | 35.21 | 81.34 | 93.57 | 90.72 | 99.82 | 99.45 | 98.90 | 95.13 | 53.12 | 56.07 | 51.06 | 93.20 | 99.63 | 91.73 | 94.90 | 95.31 | 86.26 |
| CIN | 36.18 | 93.11 | 98.90 | 99.17 | 99.91 | 100 | 99.91 | 99.72 | 86.95 | 99.72 | 98.90 | 93.57 | 99.86 | 97.66 | 99.59 | 99.59 | 97.77 |
| LM | 36.28 | 99.80 | 99.90 | 100 | 100 | 100 | 100 | 83.98 | 50.88 | 60.30 | 49.76 | 49.80 | 100 | 95.36 | 99.95 | 100 | 85.98 |
| VideoSeal | 38.82 | 98.25 | 98.07 | 98.53 | 96.51 | 98.62 | 97.61 | 99.45 | 98.16 | 99.17 | 99.45 | 96.97 | 100 | 96.65 | 99.77 | 100 | 98.48 |
| ChunkySeal | 37.12 | 88.97 | 99.82 | 99.91 | 99.45 | 99.82 | 99.54 | 98.71 | 98.25 | 86.81 | 96.42 | 99.26 | 99.91 | 96.42 | 99.59 | 99.95 | 97.52 |
| Ours | **39.26** | 97.95 | 100 | 99.80 | 100 | 99.32 | 98.93 | 99.61 | 99.22 | 100 | 100 | 99.22 | 100 | 99.81 | 100 | 100 | **99.59** |

# 5. Experiments

In this section, we show that OrthoMark resolves the capacity bottleneck in deep watermarking and sustains near-perfect decoding accuracy across a wide range of payloads. We further demonstrate that OrthoMark also improves robustness and visual quality. In all experiments, the host images have resolution $3 \times 128 \times 128$. For the capacity study, the payload ranges from 64 to 32,768 bits, doubling at each step. For the visual quality and robustness evaluations, we fix the message length to 64 bits and consider a broad suite of distortions spanning three categories and 15 subtypes. More experimental details are provided in Appendix B.

## 5.1. Decoding Accuracy under Different Payloads

As discussed in Section 1, existing deep watermarking models share a fundamental failure mode: even in a distortion-free setting, decoding accuracy decreases sharply as the payload increases. To illustrate this phenomenon, we evaluate 9 state-of-the-art (SOTA) deep watermarking baselines under varying payloads and report their bit decoding accuracy on undistorted watermarked images. For our method, we train a single model at a payload of 32,768 bits. At the inference stage, adapting to different payloads does not require retraining; we only update the orthogonal projection matrix $P$. In contrast, most baselines do not offer such flexibility, so for fairness we retrain each baseline separately for every payload. Two exceptions are ChunkySeal and SSL: ChunkySeal

cannot be scaled further due to its large parameter footprint, and SSL is constrained by its model structure, preventing scaling to higher payloads. As shown in Fig. 1, all baselines exhibit a pronounced accuracy collapse in the high capacity regime, whereas our method sustains near-perfect decoding accuracy across the evaluated range. Notably, even without further scaling, ChunkySeal already shows a declining trend at 1,024 bits, while our method remains stable.

## 5.2. Visual Quality and Robustness against Distortions

Fig. 4 visualizes the 15 distortions used in our evaluation, and Fig. 3 compares watermarked images from different methods. Table 1 reports visual quality and robustness at 64 bits. OrthoMark achieves both the highest PSNR (39.26 dB) and the best average accuracy (99.59%). It remains consistently strong across all three distortion categories, and is particularly robust under geometric transforms, where several deep watermarking baselines show noticeable gaps despite achieving comparable PSNR. While some baselines reach near-perfect accuracy on narrow subsets (e.g., LM and CIN on signal distortions), their robustness does not transfer uniformly to other categories, indicating that robustness is not a single scalar property. Table 2 reports the same comparison at a larger payload of 256 bits, with all methods evaluated at PSNR = 40 dB. At this scale, OrthoMark's advantage becomes more pronounced: it reaches an average accuracy of 90.60%, outperforming the strongest baseline (CIN) by 5.98 percentage points. The gap is espe-

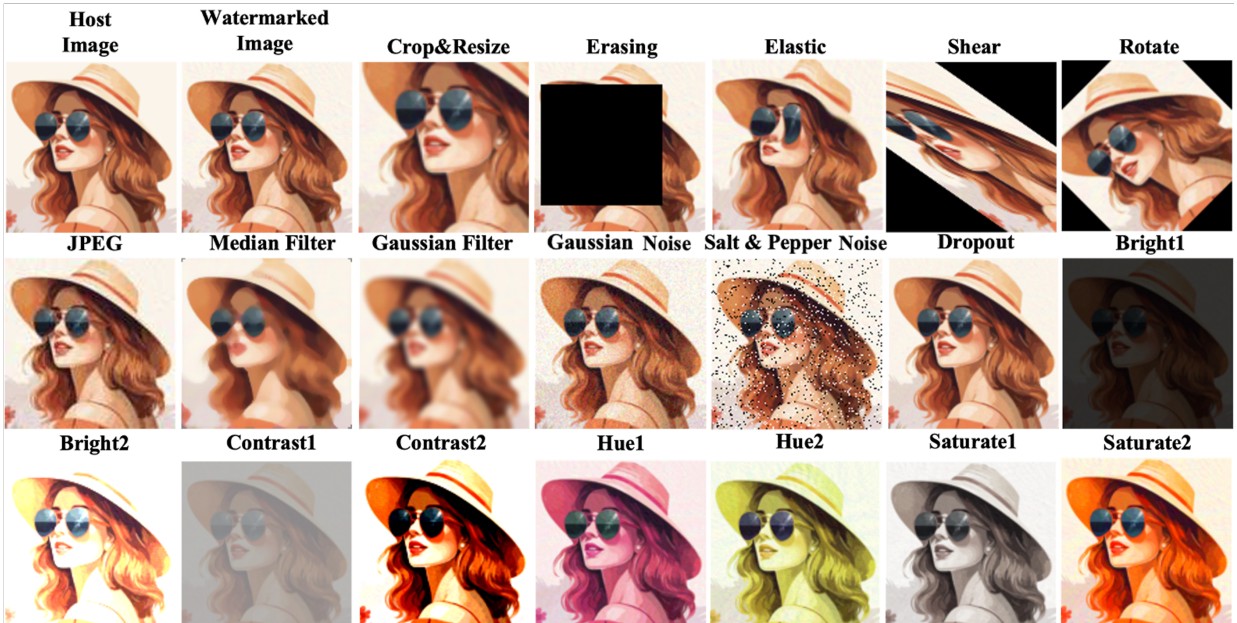

*Figure 4.* **Distortion suite used in our experiments.** We show a host image, a watermarked image, and examples of the 15 distortions spanning signal distortions, geometric transforms, and photometric transforms.

cially clear under geometric transforms, where OrthoMark maintains 75.54%, 86.58%, 87.60%, and 79.34% accuracy on crop-and-resize, shear, rotation, and elastic deformation, respectively. This supports our central claim: at 64 bits, inter-bit interference is limited and strong baselines remain competitive; at 256 bits, interference accumulates and the robustness gap widens.

## 5.3. Adjustable Visual Quality and Robustness under Single Distortions

Section 5.2 focused on the most challenging setting, where a watermarked image may encounter all 15 distortions. In practice, applications often have specific distortion preferences: users may want stronger robustness against certain distortions in exchange for improved visual quality on others. Most prior deep watermarking methods cannot accommodate this without retraining, since their encoding patterns are fixed after training. OrthoMark is trained once with a diverse set of noise layers to learn a generic robust feature domain; the inference-stage embedding can then be steered by changing the composition and strength of noise layers used during watermark generation, with no retraining. We examine this in Appendix C.3, which reports two single-distortion evaluations using the same OrthoMark checkpoint from Table 1: (i) Table 12, where OrthoMark adapts to each target distortion by changing only the inference-time noise composition (all methods use their original checkpoints; no retraining); (ii) a more in-depth setting where each baseline is fine-tuned for 10 epochs per distortion. Across both, OrthoMark remains consistently strong while enabling inference-stage

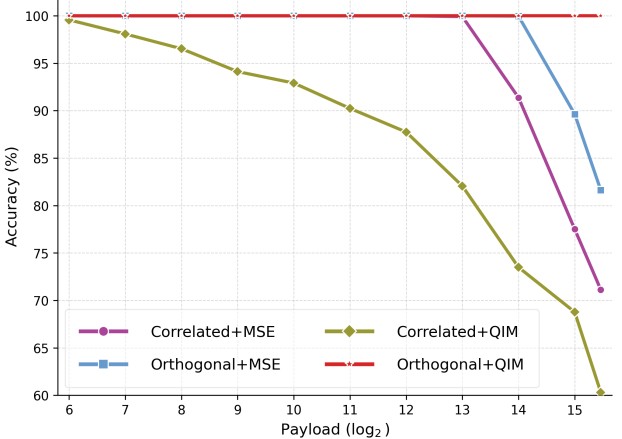

*Figure 5.* **Ablation on payload scaling in different SED module settings. (PSNR is set to 50 dB for all methods.)** Bit decoding accuracy (%) on clean watermarked images as the payload increases (reported as $\log_2$ of the payload dimension), comparing four SED variants that combine an orthogonal vs. correlated projection matrix with a QIM-style vs. MSE loss.

control of the visual quality–robustness trade-off.

## 5.4. Ablation Study

**Impact of the Orthogonal Projection Matrix and QIM-style Loss on Capacity.** Fig. 5 ablates the SED module by combining (i) an orthogonal vs. correlated projection matrix and (ii) a QIM-style loss vs. an MSE loss; orthogonal+QIM is our default OrthoMark setting. For the correlated variant, we replace the orthogonal projection matrix

*Table 2.* Benchmark comparisons at 256-bit payload under the same 15 distortions as Table 1.

| Method | PSNR (dB) | Signal Distortions | | | | | | Geometric Transforms | | | | | Photometric Transforms | | | | AVG |
|---|---|---|---|---|---|---|---|---|---|---|---|---|---|---|---|---|---|
| | | JPEG | MF | GF | DP | S&P | GN | Erase | C&R | Shear | Rotate | Elastic | Hue | Bright | Contrast | Saturate | |
| HiDDeN | 40.00 | 50.22 | 50.82 | 49.57 | 55.47 | 58.05 | 52.39 | 63.73 | 55.37 | 56.32 | 53.37 | 57.84 | 57.66 | 58.98 | 58.10 | 58.84 | 55.78 |
| LISO | 40.00 | 53.50 | 57.12 | 59.30 | 96.57 | 92.34 | 90.18 | 95.72 | 49.66 | 67.01 | 67.71 | 53.06 | 96.42 | 85.37 | 91.95 | 93.70 | 76.64 |
| SSL | 40.00 | 53.47 | 54.90 | 54.67 | 56.32 | 49.26 | 53.56 | 51.39 | 51.13 | 52.19 | 53.05 | 52.41 | 61.63 | 56.05 | 58.07 | 59.49 | 54.51 |
| FIN | 40.00 | 59.07 | 62.80 | 61.94 | 70.59 | 67.97 | 70.25 | 64.11 | 48.39 | 52.59 | 49.97 | 49.83 | 70.84 | 68.42 | 70.43 | 70.93 | 62.54 |
| MBRS | 40.00 | 53.47 | 75.07 | 81.21 | 88.30 | 91.80 | 79.71 | 77.13 | 50.35 | 57.51 | 54.32 | 58.22 | 92.10 | 78.82 | 85.42 | 88.87 | 74.15 |
| CIN | 40.00 | 67.38 | 83.25 | 92.73 | 99.87 | 98.89 | 98.09 | 94.86 | 50.22 | 74.20 | 63.44 | 51.43 | 99.98 | 95.78 | 99.35 | 99.86 | 84.62 |
| LM | 40.00 | 60.03 | 67.66 | 70.90 | 71.53 | 71.98 | 71.96 | 67.17 | 48.46 | 58.83 | 49.91 | 63.35 | 71.61 | 69.15 | 71.51 | 71.33 | 65.69 |
| VideoSeal | 40.00 | 62.83 | 69.25 | 92.43 | 91.28 | 72.44 | 72.01 | 78.15 | 96.20 | 51.14 | 50.02 | 50.65 | 94.06 | 85.48 | 93.29 | 95.94 | 77.01 |
| ChunkySeal | 40.00 | 56.64 | 69.99 | 70.99 | 71.46 | 70.94 | 68.66 | 71.38 | 72.18 | 71.23 | 68.00 | 64.93 | 72.18 | 69.88 | 72.34 | 72.04 | 69.52 |
| Ours | 40.00 | 82.36 | 96.88 | 87.37 | 97.31 | 93.25 | 84.01 | 93.42 | 75.54 | 86.58 | 87.60 | 79.34 | 99.36 | 97.25 | 98.97 | 99.71 | **90.60** |

*Table 3.* Ablation of the SED design on visual quality and robustness. "Cor" denotes the correlated carrier construction defined in Sec. 5.4, and "Oth" denotes an orthogonal projection matrix.

| Variant | PSNR (dB) | AVG |
|---|---|---|
| Cor + MSE | 32.41 | 93.82 |
| Cor + QIM | 33.61 | 92.02 |
| Oth + MSE | 39.20 | 99.45 |
| Oth + QIM (ours) | **39.26** | **99.59** |

with a correlated carrier matrix $P_{cor} = [v_1, \ldots, v_n]$, where $v_i = \rho u + (1 - \rho)\epsilon_i$. Here, $u$ is a shared direction and $\{\epsilon_i\}_{i=1}^n$ is an orthonormal basis; therefore, $\rho$ controls the amount of shared component and hence the inter-carrier correlation. First, orthogonal projection consistently improves capacity scaling, showing that enforcing independent bit channels is critical to mitigating bit-wise interference at high payloads. Second, the QIM-style loss is necessary to sustain near-perfect decoding at large payloads: MSE-based variants degrade in the high capacity regime even with orthogonal projection, whereas orthogonal+QIM maintains 100% accuracy throughout. Overall, orthogonal projection provides interference suppression by construction, and the QIM-style loss supplies the non-linear coding geometry needed for scalable capacity.

**Impact of the Orthogonal Projection Matrix and QIM-style Loss.** Table 3 ablates the SED module. First, switching from a correlated to an orthogonal projection yields a large gain in both PSNR and average accuracy, indicating that enforcing independent bit channels is crucial to suppress interference. Second, under orthogonal projection, the QIM-style loss further improves PSNR over MSE, whereas under correlated projection it does not improve average accuracy, suggesting that structured quantization is most effective when the channel geometry is properly separated. Overall, orthogonal projection is the primary driver of the visual quality–robustness gain, and the QIM-style loss provides additional refinement when paired with orthogonalized channels. The complete experimental results are provided in Table 6 from Appendix C.2.

*Table 4.* Impact of optimization iterations on visual quality and robustness.

| Metric | 500 | 1000 | 1500 | 2000 | 2500 | 3000 | 3500 | 4000 | 4500 |
|---|---|---|---|---|---|---|---|---|---|
| PSNR (dB) | 37.40 | 37.97 | 38.70 | 38.75 | 39.05 | 39.26 | 39.60 | 39.64 | **39.71** |
| AVG | 98.83 | 99.45 | 99.43 | 99.47 | 99.51 | **99.59** | 99.56 | 99.50 | 99.47 |

**Impact of Optimization Iterations on Visual Quality and Robustness.** Table 4 shows that increasing the inference-stage optimization iterations mainly improves visual quality, while robustness quickly saturates. Specifically, PSNR rises steadily from 37.40 dB at 500 iterations to 39.71 dB at 4500 iterations, with diminishing gains beyond roughly 3000 iterations. In contrast, average accuracy increases sharply from 98.83 to 99.45 when moving from 500 to 1000 iterations and then remains nearly constant across the remaining settings, peaking at 99.59 at 3000 iterations. Overall, 3000 iterations provides the best balance, achieving the highest average accuracy with strong PSNR, while additional iterations yield only marginal PSNR improvements without further robustness gains. The complete experimental results are provided in Table 7 from Appendix C.2.

## 6. Conclusion

We revisited the long-standing capacity bottleneck in deep watermarking and identified that existing methods exhibit a sharp performance collapse as the payload increases. We traced this fundamental failure mode to coding crosstalk, where bits are encoded through overlapping (non-orthogonal) carriers. To address this limitation, we proposed OrthoMark, a unified framework that combines the robustness benefits of deep watermarking with the high capacity advantages of classical coding-based watermarking. OrthoMark first learns a distortion-invariant robust feature via a Robust Feature Extractor, and then performs watermark encoding and decoding using a Structured Encoding and Decoding module with an orthogonal projection matrix and QIM-style loss. Extensive experiments demonstrate that OrthoMark significantly improves the joint trade-off among visual quality, robustness, and capacity.

## Acknowledgements

This research is supported by the National Research Foundation, Singapore under its AI Singapore Programme (AISG Award No: AISG3-RP-2022-029).

## Impact Statement

This paper introduces OrthoMark, a high capacity and robust image watermarking framework designed to address the payload bottleneck in learning-based watermarking. Our work improves the practical applicability of digital watermarking in real-world settings where reliable content attribution and provenance tracking are important, including online media distribution, copyright protection, and governance of AI-generated visual content. By enabling substantially larger payloads while maintaining strong robustness and visual quality, OrthoMark can support richer embedded metadata, such as ownership information, provenance records, and forensic identifiers, thereby helping downstream platforms verify content origin and mitigate unauthorized content distribution. Overall, this work advances practical high capacity watermarking solutions and strengthens the technical foundation for trustworthy digital media ecosystems. While our research may have broader societal implications, we do not identify any specific ethical concerns that require further discussion.

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

# A. Proof Results

## A.1. Gaussian approximation of $\Gamma_i^{\mathrm{opt}}$

Recall that

$$\Gamma_i^{\mathrm{opt}} = d_{\max}\alpha_{\min}^i \sum_{k\neq i} M_k,$$

where $\{M_k\}_{k=1}^n$ are i.i.d. Rademacher random variables, i.e., $\mathbb{P}(M_k = 1) = \mathbb{P}(M_k = -1) = 1/2$. Hence $\mathbb{E}[M_k] = 0$ and $\mathrm{Var}(M_k) = 1$.

Let $S_i \triangleq \sum_{k\neq i} M_k$. By the central limit theorem (CLT),

$$\frac{S_i - \mathbb{E}[S_i]}{\sqrt{\mathrm{Var}(S_i)}} \Rightarrow \mathcal{N}(0,1) \qquad \text{as } n \to \infty.$$

Using $\mathbb{E}[S_i] = 0$ and $\mathrm{Var}(S_i) = \sum_{k\neq i} \mathrm{Var}(M_k) = n - 1$ (by independence), we obtain

$$\frac{S_i}{\sqrt{n-1}} \Rightarrow \mathcal{N}(0,1), \qquad \text{equivalently} \qquad S_i \Rightarrow \mathcal{N}(0, n-1).$$

By scaling,

$$\Gamma_i^{\mathrm{opt}} = d_{\max}\alpha_{\min}^i S_i \Rightarrow \mathcal{N}\Big(0, \ \big(d_{\max}\alpha_{\min}^i\big)^2 (n-1)\Big).$$

Therefore, for sufficiently large $n$, $\Gamma_i^{\mathrm{opt}}$ is well approximated by a zero-mean Gaussian with variance $\big(d_{\max}\alpha_{\min}^i\big)^2 (n-1)$.

## A.2. Proof of Proposition 3.3

Under the optimistic interference setting, we have

$$y_i' = \hat{d}_i M_i \alpha_{ii} + \Gamma_i^{\mathrm{opt}}, \tag{21}$$

where $\Gamma_i^{\mathrm{opt}}$ is independent of $M_i$ and is approximated as $\Gamma_i^{\mathrm{opt}} \sim \mathcal{N}\big(0, \mathrm{Var}(\Gamma_i^{\mathrm{opt}})\big)$ by the central limit theorem (CLT). Since $M_i \in \{-1, +1\}$ and the decoder uses $\hat{M}_i = \mathrm{sign}(y_i')$, the bit error event satisfies

$$\{\hat{M}_i \neq M_i\} = \{M_i y_i' < 0\}.$$

Multiplying Equation (21) by $M_i$ gives

$$M_i y_i = \hat{d}_i \alpha_{ii} + M_i \Gamma_i^{\mathrm{opt}}.$$

Because $\Gamma_i^{\mathrm{opt}}$ is zero-mean Gaussian and independent of $M_i$, the random variable $M_i \Gamma_i^{\mathrm{opt}}$ has the same distribution as $\Gamma_i^{\mathrm{opt}}$, i.e., $M_i \Gamma_i^{\mathrm{opt}} \sim \mathcal{N}\big(0, \mathrm{Var}(\Gamma_i^{\mathrm{opt}})\big)$. Therefore,

$$P_{e,i} = \mathbb{P}(M_i y_i' < 0) = \mathbb{P}\Big(\hat{d}_i \alpha_{ii} + \Gamma_i^{\mathrm{opt}} < 0\Big)$$

$$= \mathbb{P}\Big(\Gamma_i^{\mathrm{opt}} < -\hat{d}_i \alpha_{ii}\Big) = Q\left(\frac{\hat{d}_i \alpha_{ii}}{\sqrt{\mathrm{Var}(\Gamma_i^{\mathrm{opt}})}}\right),$$

where $Q(x) = \frac{1}{\sqrt{2\pi}} \int_x^\infty e^{-u^2/2} \, \mathrm{d}u$. By the definition of $\mathrm{SIR}_i$ in Equation (10), we have

$$\mathrm{SIR}_i \triangleq \frac{(\hat{d}_i \alpha_{ii})^2}{\mathrm{Var}(\Gamma_i^{\mathrm{opt}})},$$

which yields

$$P_{e,i} = Q\Big(\sqrt{\mathrm{SIR}_i}\Big).$$

Finally, since $Q(0) = \frac{1}{2}$, $\mathrm{SIR}_i \to 0$ implies $P_{e,i} \to \frac{1}{2}$.

**A.3. SIR perspective on classical watermarking**

We now revisit classical watermarking methods through the linear superposition model. Our goal is to understand how their structured encoding and decoding mechanisms control the per-bit signal-to-interference ratio $\mathrm{SIR}_i$ in (10). By Proposition 3.3, under a fixed encoding strength, reliable decoding requires a large $\mathrm{SIR}_i$, since $P_{e,i} = Q\big(\sqrt{\mathrm{SIR}_i}\big)$ decreases as $\mathrm{SIR}_i$ increases. Therefore, classical designs aim to:

- **Maximize the useful term.** Increase the encoder–decoder alignment $\alpha_{ii} \triangleq \langle \mathbf{u}_i, \mathbf{u}_i' \rangle$.

- **Minimize the interference term.** Reduce cross-bit correlations $|\alpha_{ki}| \triangleq |\langle \mathbf{u}_k, \mathbf{u}_i' \rangle|$ for $k \neq i$, so that $\mathrm{Var}(\Gamma_i)$ remains small as the payload $n$ grows.

We will reuse this SIR-based viewpoint across LSB and QIM.

**A.3.1. LSB**

Least significant bit (LSB) (Van Schyndel et al., 1994) modulation encodes bits by overwriting the least significant bit(s) of selected pixels. A common implementation writes different bits into different pixel locations (or non-overlapping pixel groups). Let $S_i \subseteq \{1, \ldots, m\}$ denote the index set of coordinates used to encode bit $i$, so encoding bit $i$ modifies only pixels in $S_i$. In the linear model, this corresponds to using the (normalized) indicator of $S_i$ as the carrier:

$$\mathbf{u}_i \triangleq \frac{\mathbf{1}_{S_i}}{\|\mathbf{1}_{S_i}\|_2}, \qquad S_k \cap S_i = \emptyset \;\; (k \neq i). \tag{22}$$

Here $\mathbf{1}_{S_i} \in \{0,1\}^m$ is the indicator vector of $S_i$, defined component-wise by

$$\mathbf{1}_{S_i}[j] \triangleq \begin{cases} 1, & j \in S_i, \\ 0, & j \notin S_i, \end{cases} \qquad \forall j \in \{1, \ldots, m\}.$$

**Useful term: aligned decoding.** Decoding reads the same coordinates, which corresponds to a matched carrier $\mathbf{u}_i' = \mathbf{u}_i$. Therefore,

$$\alpha_{ii} = \langle \mathbf{u}_i, \mathbf{u}_i' \rangle = \langle \mathbf{u}_i, \mathbf{u}_i \rangle = 1, \tag{23}$$

so the useful term in (10) is maximized under matched decoding.

**Interference term: disjoint support.** Because different bits are written into disjoint coordinate sets, the carriers have disjoint support and are orthogonal. For any $k \neq i$,

$$\alpha_{ki} = \langle \mathbf{u}_k, \mathbf{u}_i' \rangle = \left\langle \frac{\mathbf{1}_{S_k}}{\|\mathbf{1}_{S_k}\|_2}, \frac{\mathbf{1}_{S_i}}{\|\mathbf{1}_{S_i}\|_2} \right\rangle = \frac{\langle \mathbf{1}_{S_k}, \mathbf{1}_{S_i} \rangle}{\|\mathbf{1}_{S_k}\|_2 \|\mathbf{1}_{S_i}\|_2} = 0. \tag{24}$$

Hence the aggregate interference vanishes in the noiseless setting, i.e., $\mathrm{Var}(\Gamma_i) = 0$, and $\mathrm{SIR}_i$ in (10) is maximized.

This explains why LSB can achieve high capacity with good visual quality in clean settings. However, because it operates at the quantization noise level, LSB is highly fragile to common processing such as re-quantization, compression, and filtering, which can flip the least significant bits and destroy the message.

**A.3.2. QIM**

Quantization index modulation (QIM) (Chen & Wornell, 2002) encodes each bit by selecting one of two quantizers (or cosets) indexed by the bit. In ST-QIM, the host image is first projected onto predefined carrier directions (e.g., via an orthogonal projection matrix), and each projected scalar is then quantized using a bit-dependent dither. Concretely, for bit $i$, let $\mathbf{u}_i$ denote the carrier direction and define the host image projection $s_i \triangleq \langle I_{\mathbf{ho}}, \mathbf{u}_i \rangle$. ST-QIM encodes $M_i \in \{0, 1\}$ by

$$\tilde{s}_i = \begin{cases} \Delta \left\lfloor \dfrac{s_i - d_0}{\Delta} \right\rceil + d_0, & M_i = 0, \\ \Delta \left\lfloor \dfrac{s_i - d_1}{\Delta} \right\rceil + d_1, & M_i = 1, \end{cases} \tag{25}$$

where $\Delta > 0$ is the quantization step size, $\lfloor \cdot \rceil$ denotes rounding to the nearest integer, and the two dithers index the quantizers, typically chosen as $d_0 = 0$ and $d_1 = \Delta/2$. The encoder then produces a watermarked image by updating the signal along $\mathbf{u}_i$ so that the resulting projection equals $\tilde{s}_i$.

**Useful term: aligned decoding.** ST-QIM decoding uses matched carriers, $\mathbf{u}'_i = \mathbf{u}_i$, and decides the bit by testing which quantizer reconstruction is closer to the observed projection. This gives $\alpha_{ii} = \langle \mathbf{u}_i, \mathbf{u}'_i \rangle = 1$, so encoder–decoder alignment is guaranteed by construction and the useful term in (10) is maximized under matched decoding.

**Interference term: suppressing cross-bit correlations.** ST-QIM further relies on structured carriers that are (approximately) orthogonal across bits (e.g., orthogonal projection matrix). In our notation, this enforces $|\alpha_{ki}| = |\langle \mathbf{u}_k, \mathbf{u}'_i \rangle| \approx 0$ for $k \neq i$, which keeps $\mathrm{Var}(\Gamma_i)$ small as the payload $n$ grows and prevents multi-bit interference from accumulating.

Overall, ST-QIM maintains a large $\mathrm{SIR}_i$ under a fixed encoding strength by (i) ensuring strong encoder–decoder alignment and (ii) explicitly controlling cross-bit correlations through structured (near-orthogonal) carrier design. This helps explain why QIM-style methods can support high capacity with good visual quality in clean or mildly distorted settings.

## B. Detailed Experimental Settings

**Datasets and settings.** All models are trained on COCO (Lin et al., 2014) and evaluated on the USC-SIPI image dataset (Viterbi, 1977). All images are RGB with spatial resolution $H \times W$, where $H = W = 128$. The flattened image has dimension $m$, where $m = 3HW$. The encoded watermark message has length $n$ bits. We use Adam (Kingma & Ba, 2015) with learning rate $10^{-4}$.

For training OrthoMark, we initialize the loss weights as $\lambda_1 = \lambda_2 = 1$. The image-update step size is $\eta = 10^{-3}$, and we run $T = 3000$ update iterations. For the structured encoding and decoding module, the quantization step size is set to $\Delta = 1$. To evaluate capacity scaling, we vary the message length from $n = 64$ to $n = 32768$, doubling $n$ each time.

For robustness training, we consider three categories comprising 15 distortions. **Signal distortions** include Gaussian blur (GF; standard deviation 2.0, kernel size 7), median filtering (MF; kernel size 7), additive Gaussian noise (GN; mean 0, variance 0.04), salt-and-pepper noise (S&P; noise ratio 0.1), dropout (DP; drop ratio 0.5), and JPEG compression (JPEG; quality factor 50). **Geometric distortions** include shear (degrees in $[-55, 55]$), rotation (degrees in $[-45, 45]$), elastic deformation (strength $\alpha = 3$), crop-and-resize (crop rate 50%), and random erasing (erase rate 50%). **Photometric distortions** include brightness adjustment (Bright; factor in $[0.2, 2.0]$), contrast adjustment (Contrast; factor in $[0.2, 2.0]$), saturation adjustment (Saturate; factor in $[0.2, 2.0]$), and hue shift (Hue; offset in $[-0.1, 0.1]$).

For robustness testing, we report decoding accuracy under the following distortion strengths. **Signal distortions** include Gaussian blur (GF; standard deviation 2.0, kernel size 7), median filtering (MF; kernel size 7), additive Gaussian noise (GN; mean 0, variance 0.04), salt-and-pepper noise (S&P; noise ratio 0.1), dropout (DP; drop ratio 0.5), and JPEG compression (JPEG; quality factor 50). **Geometric distortions** include shear (we average the results at $-55°$ and $+55°$), rotation (we average the results at $-45°$ and $+45°$), elastic deformation (strength $\alpha = 3$), crop-and-resize (crop rate 50%), and random erasing (erase rate 50%). **Photometric distortions** include brightness adjustment (Bright; we average the results at factors 0.2 and 2.0), contrast adjustment (Contrast; we average the results at factors 0.2 and 2.0), saturation adjustment (Saturate; we average the results at factors 0.2 and 2.0), and hue shift (Hue; we average the results at offsets $-0.1$ and $+0.1$).

All experiments are implemented in PyTorch (Paszke et al., 2019) and run on an NVIDIA RTX A40 GPU.

**Baselines** We compare OrthoMark against 10 representative baselines spanning both classical and deep-learning-based watermarking methods, including DWT–DCT–SVD (Navas et al., 2008), HiDDeN (Zhu et al., 2018), MBRS (Jia et al., 2021), LISO (Chen et al., 2022), SSL (Fernandez et al., 2022), CIN (Ma et al., 2022), FIN (Fang et al., 2023), VideoSeal (Fernandez et al., 2024), LM (Qiu et al., 2025), and ChunkySeal (Petrov et al., 2025).

**Evaluation metrics.** We evaluate visual quality using peak signal-to-noise ratio (PSNR), where a higher PSNR indicates that the watermarked image is closer to the host image. We evaluate robustness using decoding accuracy under distortions, where higher accuracy indicates stronger robustness.

*Table 5.* **Plug-in ablation with SED.** For each baseline, "(SED)" denotes replacing the baseline's original encoding procedure with our Structured Encoding and Decoding (SED) and generating watermarked images via iterative optimization, while keeping the baseline decoder fixed. We report visual quality and decoding accuracy under the same distortion suite as in Table 1.

| Method | PSNR (dB) | Signal Distortions | | | | | | Geometric Transforms | | | | | Photometric Transforms | | | | AVG |
|---|---|---|---|---|---|---|---|---|---|---|---|---|---|---|---|---|---|
| | | JPEG | MF | GF | DP | S&P | GN | Erase | C&R | Shear | Rotate | Elastic | Hue | Bright | Contrast | Saturate | |
| HiDDeN | 33.60 | 56.99 | 51.10 | 53.03 | 65.90 | 61.86 | 57.90 | 70.13 | 64.06 | 58.27 | 55.97 | 67.00 | 68.80 | 64.11 | 69.12 | 72.06 | 62.42 |
| HiDDeN(SED) | 35.59 | 52.48 | 94.39 | 52.21 | 96.60 | 62.68 | 62.87 | 94.67 | 90.07 | 97.98 | 98.62 | 95.68 | 99.77 | 98.39 | 98.62 | 99.63 | 86.31 |
| LISO | 33.04 | 53.54 | 62.04 | 67.83 | 98.30 | 97.98 | 98.44 | 97.98 | 50.92 | 48.67 | 49.93 | 62.41 | 96.81 | 96.48 | 99.68 | 99.31 | 78.69 |
| LISO(SED) | 31.32 | 52.07 | 100 | 56.94 | 51.47 | 50.78 | 48.71 | 59.47 | 50.74 | 99.66 | 99.89 | 51.75 | 91.36 | 85.62 | 85.66 | 85.96 | 71.34 |
| SSL | 35.12 | 63.60 | 68.57 | 70.22 | 73.16 | 49.26 | 67.74 | 54.60 | 59.19 | 55.74 | 68.11 | 59.93 | 79.27 | 73.58 | 78.35 | 78.17 | 66.63 |
| SSL(SED) | 37.58 | 50.92 | 58.55 | 89.61 | 87.32 | 49.91 | 48.44 | 51.84 | 50.37 | 89.25 | 93.24 | 50.74 | 78.68 | 88.14 | 85.57 | 84.24 | 70.45 |
| FIN | 35.74 | 95.21 | 92.87 | 95.70 | 99.61 | 87.11 | 99.80 | 89.06 | 52.73 | 69.43 | 50.05 | 70.90 | 100 | 94.14 | 98.73 | 100 | 86.36 |
| FIN(SED) | 35.75 | 79.59 | 99.22 | 49.32 | 99.90 | 53.81 | 99.71 | 75.68 | 49.41 | 91.06 | 93.85 | 50.00 | 100 | 97.31 | 99.07 | 100 | 82.53 |
| MBRS | 35.21 | 81.34 | 93.57 | 90.72 | 99.82 | 99.45 | 98.90 | 95.13 | 53.12 | 56.07 | 51.06 | 93.20 | 99.63 | 91.73 | 94.90 | 95.31 | 86.26 |
| MBRS(SED) | 36.58 | 70.68 | 99.54 | 78.31 | 100 | 99.45 | 99.91 | 90.53 | 53.95 | 98.76 | 95.54 | 74.82 | 100 | 98.30 | 99.82 | 99.86 | 90.63 |
| CIN | 36.18 | 93.11 | 98.90 | 99.17 | 99.91 | 100 | 99.91 | 99.72 | 86.95 | 99.72 | 98.90 | 93.57 | 99.86 | 97.66 | 99.59 | 99.59 | 97.77 |
| CIN(SED) | 37.41 | 86.33 | 97.75 | 95.21 | 97.36 | 96.97 | 97.36 | 96.00 | 58.79 | 95.95 | 96.97 | 76.56 | 98.24 | 97.17 | 97.61 | 97.66 | 92.40 |
| LM | 36.28 | 99.80 | 99.90 | 100 | 100 | 100 | 100 | 83.98 | 50.88 | 60.30 | 49.76 | 49.80 | 100 | 95.36 | 99.95 | 100 | 85.98 |
| LM(SED) | 36.85 | 93.16 | 96.00 | 52.15 | 87.79 | 86.13 | 54.79 | 70.90 | 51.86 | 85.84 | 84.03 | 52.05 | 99.61 | 92.04 | 83.30 | 98.19 | 79.19 |
| VideoSeal | 38.82 | 98.25 | 98.07 | 98.53 | 96.51 | 98.62 | 97.61 | 99.45 | 98.16 | 99.17 | 99.45 | 96.97 | 100 | 96.65 | 99.77 | 100 | 98.48 |
| VideoSeal(SED) | 27.42 | 51.56 | 70.22 | 86.76 | 48.53 | 47.61 | 52.02 | 48.35 | 76.38 | 87.55 | 85.20 | 50.92 | 85.62 | 81.71 | 81.57 | 77.76 | 68.78 |
| ChunkySeal | 37.12 | 88.97 | 99.82 | 99.91 | 99.45 | 99.82 | 99.54 | 98.71 | 98.25 | 86.81 | 96.42 | 99.26 | 99.91 | 96.42 | 99.59 | 99.95 | 97.52 |
| ChunkySeal(SED) | 24.22 | 57.17 | 85.57 | 90.81 | 67.74 | 73.81 | 77.30 | 53.86 | 91.54 | 92.78 | 92.19 | 56.99 | 91.87 | 93.89 | 92.37 | 90.30 | 80.55 |
| Ours | **39.26** | 97.95 | 100 | 99.80 | 100 | 99.32 | 98.93 | 99.61 | 99.22 | 100 | 100 | 99.22 | 100 | 99.81 | 100 | 100 | **99.59** |

# C. Extensive Experimental Results

## C.1. Ablation: Plugging SED into Existing Deep Watermarking Decoders

To isolate the contribution of our structured coding module, we conduct an ablation where we decouple the decoder of each deep watermarking baseline trained in Sec. 5.2 and replace its original encoding and decoding procedure with our Structured Encoding and Decoding (SED) module. Concretely, we keep each baseline decoder fixed, generate watermarked images via our iterative optimization with SED, and then decode using the corresponding baseline decoder. This plug-in setting (denoted as "(SED)") evaluates whether structured coding and orthogonalized channels can improve robustness and visual quality independently of the specific decoder architecture.

As shown in Table 5, we conclude that:

**Overall trend.** For most baselines, the "(SED)" variant substantially improves the joint performance, often increasing PSNR and lifting the overall average accuracy, which indicates that a non-trivial part of the robustness and visual quality bottleneck comes from the encoding side (i.e., how bits are written into the representation), rather than solely from the decoder architecture.

**Large gains when the original encoder is the bottleneck.** Several decoders benefit dramatically from SED. For example, HiDDeN(SED) improves average accuracy from 62.42 to 86.31 while increasing PSNR, and MBRS(SED) improves average accuracy from 86.26 to 90.63. These gains suggest that structured coding with orthogonalized channels can provide a substantially cleaner and more decodable watermark signal for the same decoder, especially under geometric and photometric transforms where many baselines struggle.

**When SED helps less.** Not all methods improve. Strong baselines that already achieve high average accuracy with their own pipeline (e.g., CIN) see smaller or even negative changes (CIN: $97.77 \rightarrow 92.40$), implying that their decoders are tuned to the statistics induced by their own encoder. In such cases, replacing the encoder with SED introduces a distribution shift that the fixed decoder is not optimized for, limiting the plug-in benefit.

**Failure cases highlight encoder–decoder co-adaptation.** We also observe clear degradations for some high-performing systems (e.g., VideoSeal and ChunkySeal), where PSNR and average accuracy drop sharply after plugging in SED. This likely reflects strong encoder–decoder co-adaptation in these pipelines: their decoders may rely on method-specific encoding

*Table 6.* Ablation of the SED design on visual quality and robustness. "Cor" denotes a correlated projection matrix, and "Oth" denotes an orthogonal projection matrix.

| Method | PSNR (dB) | Signal Distortions | | | | | | Geometric Transforms | | | | | Photometric Transforms | | | | AVG |
|---|---|---|---|---|---|---|---|---|---|---|---|---|---|---|---|---|---|
| | | JPEG | MF | GF | DP | S&P | GN | Erase | C&R | Shear | Rotate | Elastic | Hue | Bright | Contrast | Saturate | |
| Cor+MSE | 32.41 | 92.01 | 97.92 | 97.57 | 99.91 | 98.35 | 99.22 | 97.14 | 77.60 | 84.59 | 82.47 | 81.42 | 100 | 99.09 | 99.96 | 100 | 93.82 |
| Cor+QIM | 33.61 | 92.36 | 73.00 | 96.70 | 99.91 | 98.96 | 99.91 | 98.70 | 77.69 | 80.21 | 76.56 | 87.93 | 99.83 | 98.74 | 99.83 | 100 | 92.02 |
| Oth+MSE | 39.20 | 98.78 | 100 | 100 | 100 | 98.52 | 99.39 | 99.83 | 98.18 | 100 | 100 | 97.05 | 100 | 100 | 100 | 100 | 99.45 |
| Oth+QIM (ours) | 39.26 | 97.95 | 100 | 99.80 | 100 | 99.32 | 98.93 | 99.61 | 99.22 | 100 | 100 | 99.22 | 100 | 99.81 | 100 | 100 | 99.59 |

*Table 7.* Impact of optimization iterations on visual quality and robustness.

| Iteration | PSNR (dB) | Signal Distortions | | | | | | Geometric Transforms | | | | | Photometric Transforms | | | | AVG |
|---|---|---|---|---|---|---|---|---|---|---|---|---|---|---|---|---|---|
| | | JPEG | MF | GF | DP | S&P | GN | Erase | C&R | Shear | Rotate | Elastic | Hue | Bright | Contrast | Saturate | |
| 500 | 37.40 | 98.24 | 99.80 | 100 | 99.90 | 98.24 | 99.61 | 99.32 | 97.85 | 94.97 | 98.93 | 96.19 | 100 | 99.37 | 100 | 100 | 98.83 |
| 1000 | 37.97 | 98.24 | 100 | 99.61 | 100 | 98.44 | 99.22 | 99.71 | 99.22 | 99.46 | 99.61 | 98.34 | 100 | 99.85 | 100 | 100 | 99.45 |
| 1500 | 38.70 | 97.56 | 100 | 100 | 100 | 98.93 | 99.22 | 99.61 | 98.83 | 99.90 | 100 | 97.66 | 100 | 99.76 | 100 | 100 | 99.43 |
| 2000 | 38.75 | 97.17 | 100 | 99.90 | 100 | 98.73 | 99.80 | 99.80 | 98.93 | 99.46 | 100 | 98.44 | 100 | 99.76 | 100 | 100 | 99.47 |
| 2500 | 39.05 | 97.56 | 100 | 100 | 100 | 99.12 | 99.22 | 99.71 | 99.22 | 99.95 | 100 | 98.05 | 100 | 99.85 | 100 | 100 | 99.51 |
| 3000 (ours) | 39.26 | 97.95 | 100 | 99.80 | 100 | 99.32 | 98.93 | 99.61 | 99.22 | 100 | 100 | 99.22 | 100 | 99.81 | 100 | 100 | 99.59 |
| 3500 | 39.60 | 97.95 | 100 | 99.85 | 100 | 98.93 | 99.05 | 100 | 99.41 | 100 | 100 | 98.44 | 100 | 99.76 | 100 | 100 | 99.56 |
| 4000 | 39.64 | 97.66 | 100 | 99.80 | 100 | 98.14 | 99.02 | 99.90 | 99.33 | 100 | 100 | 98.85 | 100 | 99.73 | 100 | 100 | 99.50 |
| 4500 | 39.71 | 97.46 | 100 | 100 | 100 | 98.02 | 99.00 | 99.80 | 99.02 | 100 | 100 | 98.83 | 100 | 99.85 | 100 | 100 | 99.47 |

artifacts or feature conventions, so changing the encoding mechanism without re-training the decoder can break the implicit contract between the two components.

**Takeaway.** These results support two conclusions. First, structured coding and orthogonalized channels are powerful and can significantly enhance robustness and visual quality for many decoders even without modifying the decoder. Second, end-to-end deep watermarking methods often entangle representation learning with watermark coding, so a plug-and-play swap is not universally effective; fully realizing the benefit of SED may require a compatible feature domain and light decoder adaptation rather than keeping the decoder entirely fixed.

## C.2. Detailed Experimental Results for Section 5.4

This appendix provides the full ablation results corresponding to Section 5.4. For the correlated carrier baseline, we replace the orthogonal projection matrix with $P_{\text{cor}} = [v_1, \ldots, v_n]$, where $v_i = \rho u + (1 - \rho)\epsilon_i$, $u$ is a shared direction, and $\{\epsilon_i\}_{i=1}^{n}$ is an orthonormal basis. Thus, $\rho$ controls the amount of shared component and the resulting inter-carrier correlation. First, Table 6 reports an ablation of the Structured Encoding and Decoding (SED) module, comparing correlated vs. orthogonal projection matrices and MSE vs. QIM-style loss functions, to quantify how channel geometry and coding objectives affect visual quality and robustness. Second, Table 7 reports the impact of the inference-stage optimization budget by varying the number of optimization iterations, which characterizes the trade-off between improved visual quality and saturated robustness as optimization iterations increase.

## C.3. Detailed Experimental Results for Section 5.3

This appendix provides the full single distortion evaluation results corresponding to Section 5.3. Table 12 first evaluates the inference-stage adjustment setting. For each distortion type, we use the checkpoints from Table 1, vary the distortion severity during inference, and report both visual quality and decoding accuracy. In this setting, OrthoMark changes only the inference-time noise composition during watermark generation; no model retraining is performed. This table demonstrates that a single OrthoMark checkpoint can be steered toward different robustness profiles.

Table 13 provides a stricter baseline-adaptation comparison. For each specific distortion family, baseline methods are fine-tuned for 10 epochs under that distortion before evaluation. This setting gives the baselines attack-specific training and complements the one-checkpoint inference-stage evaluation above. The results show that fine-tuning improves several baselines on some signal and photometric distortions. For example, LM-ft and CIN-ft reach near-perfect accuracy on Gaussian noise, dropout, blur, and several color transformations. However, the gains are not uniform across distortion

families or strengths. Under geometric transforms, several fine-tuned baselines still degrade sharply at stronger settings: FIN-ft and LM-ft remain close to chance under large rotations, VideoSeal-ft drops to $49.86\%$ at $135°$ rotation and $60.25\%$ at shear strength 75, and MBRS-ft drops to $55.38\%$ at the same shear setting. Crop-and-resize also remains difficult for several baselines after fine-tuning, with FIN-ft and LM-ft reaching only $66.93\%$ and $61.11\%$ at the strongest setting. In contrast, OrthoMark maintains consistently high accuracy across the same distortions, including $100\%$ accuracy for all rotation strengths and at least $98.01\%$ on crop-and-resize, while also preserving strong visual quality. This indicates that the advantage is not solely due to assuming a target distortion at inference time; even when baselines receive attack-specific fine-tuning, OrthoMark retains a more stable robustness profile.

*Table 8.* Latency comparison (ms/img). Encoding measures watermark generation only; decoding measures message extraction on prebuilt inputs.

| Method | Encoding latency (ms/img) | Decoding latency (ms/img) |
|---|---|---|
| LM | 0.111 | 0.094 |
| VideoSeal | 0.764 | 0.668 |
| FIN | 1.107 | 1.094 |
| MBRS | 2.248 | 0.950 |
| CIN | 4.244 | 4.589 |
| ChunkySeal | 3.439 | 3.692 |
| SSL | 3391.483 | 0.297 |
| LISO | 7124.736 | 0.385 |
| ResNet (T=600) | 2282.315 | 1.380 |
| OrthoMark (T=600) | **1151.212** | **0.887** |

## C.4. Encoding Latency and Practical Use Cases

The encoding stage of OrthoMark is optimization-based: a watermarked image is generated by iteratively updating the input image so that its robust feature representation matches the structured encoded target. This design introduces an encoding-time cost, but it also enables a substantially higher-capacity operating regime. For comparison, ChunkySeal scales VideoSeal's embedder by $90\times$ and extractor by $23\times$, while increasing the payload from 256 to 1024 bits. OrthoMark instead targets 32,768 bits by decoupling robust feature extraction from structured message encoding. Thus, the method represents a different point in the design space, trading additional encoding computation for high payload, robustness, and visual quality.

The optimization budget is controlled by the number of inference-stage update steps $T$. Table 7 shows that performance exhibits diminishing returns as $T$ increases. While the main experiments use $T = 3000$, Table 7 shows that even $T = 500$ already surpasses all baselines in Table 1 in average accuracy. Table 8 therefore reports latency under a practical lower-cost setting with $T = 600$. All latency measurements are obtained on an NVIDIA A100 GPU using $128 \times 128$ RGB images. Among optimization-based methods, OrthoMark has the lowest encoding latency in the comparison: it is $2.9\times$ faster than SSL and $6.2\times$ faster than LISO. OrthoMark's decoder is single-pass and runs in $0.887$ ms/image, which is comparable to FIN and MBRS.

This latency profile is most relevant to workflows in which robust high capacity embedding is performed offline and decoding is performed frequently. One example is media provenance certification, where a photograph may be signed once before long-term redistribution and later verified many times by downstream platforms. In this setting, the payload can store structured provenance metadata, while the main throughput requirement is fast verification. Another example is generative AI content governance, where forensic metadata for attribution or abuse investigation may require both large payload and robustness to downstream transformations. In such cases, an offline encoding cost on the order of one second can be acceptable when paired with single-pass decoding.

*Table 9.* Backbone and DWT ablation under the 15 noises. *w/o DWT*: INN backbone without DWT preprocessing. *ResNet*: ResNet backbone without DWT. Ours: INN backbone with DWT preprocessing.

| Method | PSNR | JPEG | MF | GF | DP | S&P | GN | Erase | C&R | Shear | Rotate | Elastic | Hue | Bright | Contrast | Saturate | AVG |
|---|---|---|---|---|---|---|---|---|---|---|---|---|---|---|---|---|---|
| w/o DWT | 39.12 | 89.32 | **100** | 99.74 | 99.91 | 96.61 | 99.48 | **99.83** | 92.71 | **100** | **100** | 89.32 | **100** | **100** | **100** | 99.91 | 97.79 |
| ResNet | 35.85 | **100** | **100** | 81.25 | 99.91 | 91.75 | **99.74** | 99.05 | 66.75 | 99.48 | 99.91 | 85.94 | 99.57 | 97.96 | 98.26 | 98.91 | 94.57 |
| **Ours** | **39.26** | 97.95 | **100** | **99.80** | **100** | **99.32** | 98.93 | 99.61 | **99.22** | **100** | **100** | **99.22** | **100** | 99.81 | **100** | **100** | **99.59** |

## C.5. Backbone Structure and DWT Ablation

This section analyzes the architectural choices in the robust feature extractor. Although OrthoMark does not call the inverse path $f^{-1}(z)$ during inference, the bijective property of the INN backbone remains important for the feature-space optimization used during encoding. For a fixed cover image, two different messages $m_1 \neq m_2$ are mapped by the QIM-style encoder to different coset targets $z_{m_1} \neq z_{m_2}$. Successful decoding requires the corresponding optimized watermarked images to remain distinguishable after feature extraction. A bijective feature extractor preserves this distinction at the architectural level: distinct feature targets correspond to distinct image-space solutions. In contrast, a non-bijective backbone can map multiple images to the same feature representation, making the optimization problem less constrained. As the payload increases and the QIM cosets become denser, such feature collisions can make distinct messages harder to separate, contributing to the capacity degradation observed with non-bijective architectures.

Table 9 evaluates this effect by replacing the INN backbone with a ResNet backbone while keeping the remaining setting unchanged. The ResNet variant achieves a substantially lower average accuracy (94.57%) than the full model (99.59%), with especially large drops under geometric distortions such as crop-and-resize and elastic deformation. As shown in Table 8, this substitution also does not provide an efficiency advantage: the ResNet variant requires 2282.315 ms/image for encoding and 1.380 ms/image for decoding, both higher than OrthoMark. This result indicates that the bijective structure of the INN provides a useful inductive bias for high capacity feature-domain watermarking, even though the reverse network path is not explicitly used at inference.

The same table also ablates the DWT preprocessing used in the INN backbone. Removing DWT reduces the average accuracy from 99.59% to 97.79%, with notable degradation under JPEG compression and elastic deformation. This suggests that the DWT decomposition helps organize low- and high-frequency information into a representation that is more robust to the distortion suite considered in this work.

*Table 10.* Generalization without retraining across image resolutions and message lengths. Columns within each metric denote message length in bits; entries marked "–" are not evaluated; the large-resolution settings use a fixed 64-bit message.

| Resolution | PSNR (dB) | | | | AVG (%) | | | |
|---|---|---|---|---|---|---|---|---|
| | 32 | 64 | 128 | 256 | 32 | 64 | 128 | 256 |
| $64 \times 64$ | 39.01 | 38.99 | 39.74 | 41.23 | 97.22 | 91.19 | 82.20 | 71.01 |
| $128 \times 128$ | 40.21 | 40.61 | 39.23 | 39.47 | 99.90 | 99.73 | 96.83 | 90.91 |
| $256 \times 256$ | 38.35 | 39.14 | 39.55 | 39.48 | 99.55 | 99.48 | 97.95 | 96.26 |
| $64 \times 128$ | 39.75 | 39.05 | 39.34 | 40.09 | 99.48 | 96.78 | 91.82 | 82.09 |
| $128 \times 256$ | 39.22 | 39.55 | 39.58 | 39.24 | 99.86 | 99.59 | 98.21 | 94.91 |
| $1024 \times 1024$ | – | 40.78 | – | – | – | 96.09 | – | – |
| $2048 \times 2048$ | – | 40.76 | – | – | – | 96.00 | – | – |
| $4096 \times 4096$ | – | 40.73 | – | – | – | 95.95 | – | – |

## C.6. Resolution Generalization

Table 10 evaluates the same OrthoMark checkpoint from Table 1 without retraining across different image resolutions and payload lengths. Across square and non-square inputs, OrthoMark maintains stable visual quality and decoding accuracy over payloads from 32 to 256 bits, showing that the learned robust feature extractor generalizes beyond the training resolution.

The same table also reports target-resolution results at $1024 \times 1024$, $2048 \times 2048$, and $4096 \times 4096$ with a fixed 64-bit payload. Performance remains effective at $4096 \times 4096$ resolution, corresponding to 16M pixels, with 40.73 dB PSNR and 95.95% average decoding accuracy. This scalability follows from the fact that OrthoMark has no explicit feed-forward encoder whose architecture must align image and message feature maps across resolutions. The projection subspace induced by the orthogonal projection matrix $P$ is training-free, so adapting to a new image size only requires constructing the corresponding projection matrix. Therefore, OrthoMark can scale to megapixel-resolution images without retraining the model.

## C.7. QIM Variant Ablation

Table 11 compares different QIM variants within the same OrthoMark framework. ST-QIM (Chen & Wornell, 2002), DC-QIM (Chen & Wornell, 2002), and the 2-D hexagonal-lattice (Moulin & Koetter, 2005) A2-QIM variant achieve similar overall performance, with average decoding accuracies of 99.59%, 99.15%, and 99.22%, respectively. This result indicates that OrthoMark is not tied to a specific QIM realization: the structured encoding module can incorporate different quantization variants.

*Table 11.* Ablation on QIM variants under the 15 distortions. DC-QIM uses $\alpha$=0.5. Bold indicates the better result per column.

| Method | PSNR | JPEG | MF | GF | DP | S&P | GN | Erase | C&R | Shear | Rotate | Elastic | Hue | Bright | Contrast | Saturate | AVG |
|---|---|---|---|---|---|---|---|---|---|---|---|---|---|---|---|---|---|
| A2-QIM | **39.42** | **98.96** | 97.31 | **100** | **100** | 99.31 | 96.35 | **100** | **100** | **100** | 98.44 | 97.92 | 99.96 | **100** | **100** | **100** | 99.22 |
| DC-QIM | 39.35 | 98.44 | 97.92 | 99.57 | 99.61 | **99.74** | 97.05 | 99.91 | 99.39 | 99.31 | 98.61 | 98.18 | 99.74 | 99.87 | 99.96 | 99.91 | 99.15 |
| ST-QIM (ours) | 39.26 | 97.95 | **100** | 99.80 | **100** | 99.32 | **98.93** | 99.61 | 99.22 | **100** | **100** | **99.22** | **100** | 99.81 | **100** | **100** | **99.59** |

For the 2-D hexagonal-lattice variant, the projected coefficients are not quantized independently. Instead, adjacent projected coefficients are grouped into 2-D blocks and jointly quantized on the hexagonal lattice $A_2$. We use the nested lattice pair $2A_2 \subset A_2$, so the quotient $A_2/2A_2$ defines four cosets corresponding to the four 2-bit patterns. For each 2-D block, the encoder searches for the nearest lattice point in the target coset and uses this point as the quantization target. The training objective is an MSE loss toward this target lattice point, or its distortion-compensated variant, rather than the coordinate-wise cosine periodic loss used by ST-QIM. At decoding time, each received 2-D block is assigned to the closest of the four cosets, yielding the corresponding 2-bit decision.

## C.8. Visualization of Learned Geometric Adaptation

To examine how end-to-end training adapts the watermark signal to geometric transforms, we train OrthoMark separately with individual geometric noise layers and visualize the resulting embedding residuals. Fig. 6 shows that the learned patterns are distortion dependent: training with rotation produces ring-shaped residuals, training with shear concentrates the signal in central regions with parallel structures, and training with random erasing shifts more embedding energy toward image borders. These structures are consistent with the spatial characteristics of the corresponding transformations, indicating that the learned robust feature domain can adapt watermark placement to geometric robustness requirements.

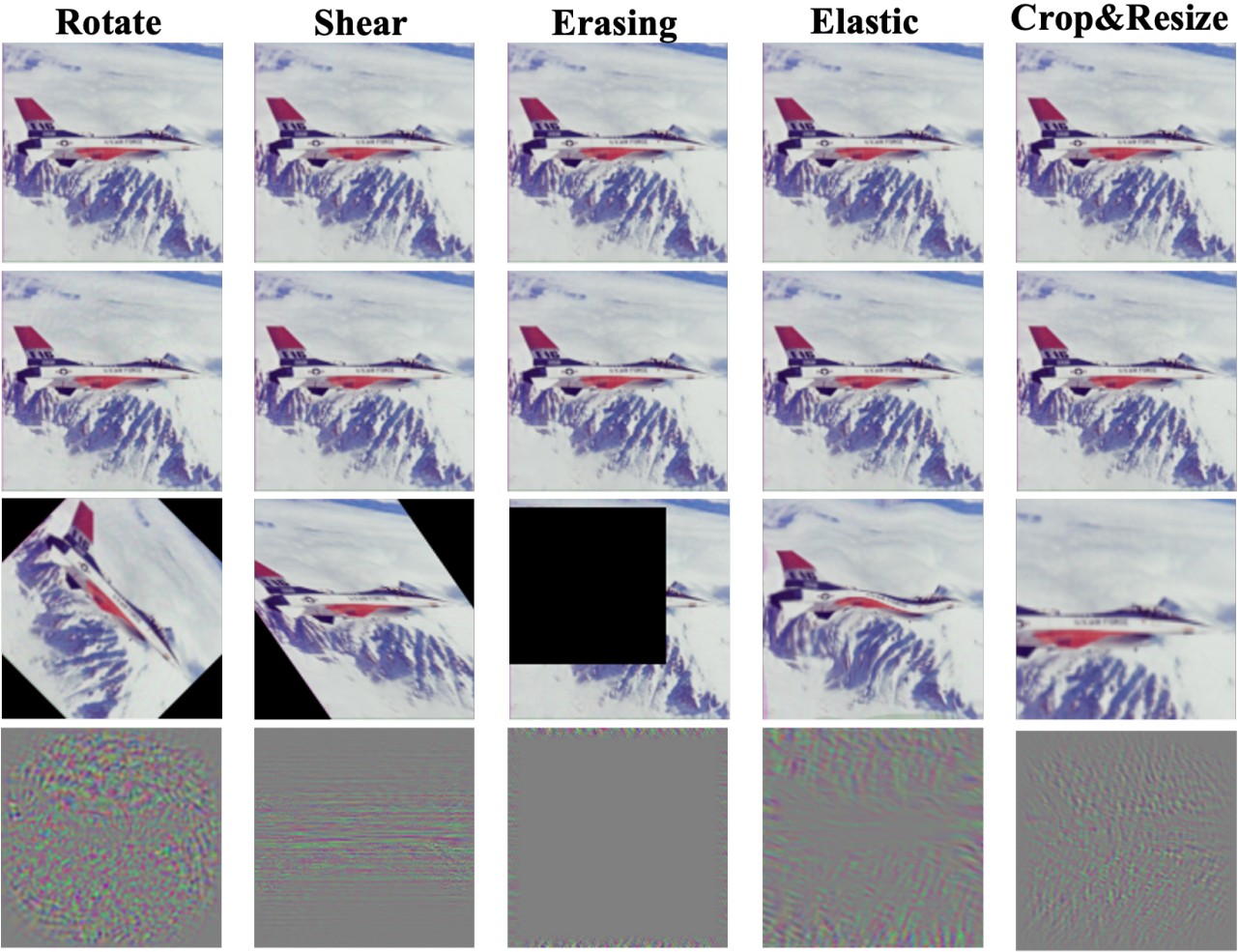

*Figure 6.* Visualization of watermark embedding patterns learned by OrthoMark when trained separately under individual geometric distortions. From top to bottom, the rows show the original images, watermarked images, distorted watermarked images, and watermark residuals amplified by $10\times$.

## C.9. Limitations and Future Work

OrthoMark trades encoding efficiency for high capacity, robustness, and visual quality. Unlike single-pass feed-forward encoders, its embedding stage is optimization-based: the watermarked image is generated by iteratively updating the input so that its robust feature representation matches the structured coding target. As shown in Table 8, this yields higher encoding latency than feed-forward methods. Table 7 also shows that the budget can be reduced in practice: even $T = 500$ exceeds all baselines in Table 1 in average accuracy, while $T = 3000$ provides better performance.

This trade-off makes OrthoMark suitable for scenarios where high capacity and robustness matter more than real-time embedding, such as offline media provenance, copyright protection for high-value visual assets, and post-hoc forensic attribution. These workflows often watermark an image once but verify it many times, making the single-pass decoder useful for high-throughput verification. Conversely, OrthoMark is not suitable for ultra-low-latency streaming or resource-constrained real-time embedding, where single-pass encoding remains preferable.

Future work includes amortizing iterative embedding into a feed-forward encoder through distillation or supervised training on optimized OrthoMark outputs. Another direction is video watermarking; this paper focuses on still images, while video requires temporal consistency and video-specific compression and motion handling.

*Table 12.* **Single-noise inference-stage evaluation.** For each method, we use the checkpoint from Table 1 without retraining and report the stego PSNR (dB) and decoding accuracy (%) under different strengths of a single distortion.

| Method | JPEG PSNR(dB) | 50 | 60 | 70 | 80 | 90 | Gaussian Noise PSNR(dB) | 0.03 | 0.04 | 0.05 | 0.06 | 0.07 | S&P PSNR(dB) | 0.02 | 0.05 | 0.10 | 0.15 | 0.20 |
|---|---|---|---|---|---|---|---|---|---|---|---|---|---|---|---|---|---|---|
| HiDDeN | 33.60 | 56.99 | 57.26 | 57.26 | 58.27 | 59.38 | 33.60 | 58.09 | 58.09 | 57.08 | 56.62 | 56.99 | 33.60 | 73.25 | 65.07 | 60.20 | 57.63 | 54.14 |
| LISO | 33.04 | 53.54 | 53.77 | 56.57 | 60.80 | 67.69 | 33.04 | 97.93 | 98.44 | 98.12 | 98.02 | 97.66 | 33.04 | 99.49 | 98.44 | 97.98 | 97.70 | 97.47 |
| SSL | 35.12 | 63.60 | 63.60 | 67.46 | 71.05 | 76.47 | 35.12 | 66.27 | 67.74 | 66.18 | 66.54 | 67.28 | 35.12 | 51.75 | 48.53 | 49.26 | 49.91 | 48.71 |
| FIN | 35.74 | 95.21 | 97.95 | 98.73 | 99.80 | 100 | 35.74 | 100 | 99.80 | 100 | 99.90 | 99.90 | 35.74 | 99.02 | 96.29 | 87.11 | 79.39 | 74.90 |
| MBRS | 35.21 | 81.34 | 83.55 | 88.33 | 90.81 | 96.05 | 35.21 | 99.08 | 99.26 | 98.53 | 98.44 | 98.90 | 35.21 | 100 | 100 | 99.54 | 98.44 | 98.07 |
| CIN | 36.18 | 93.11 | 94.58 | 96.23 | 97.89 | 99.54 | 36.18 | 100 | 99.91 | 99.63 | 99.17 | 97.70 | 36.18 | 100 | 100 | 100 | 99.91 | 99.45 |
| LM | 36.28 | 99.80 | 100 | 100 | 100 | 100 | 36.28 | 100 | 100 | 100 | 100 | 100 | 36.28 | 100 | 100 | 100 | 100 | 99.90 |
| VideoSeal | 38.82 | 98.25 | 99.08 | 99.72 | 99.54 | 100 | 38.82 | 98.81 | 98.25 | 95.68 | 94.39 | 93.01 | 38.82 | 99.91 | 99.91 | 99.08 | 96.32 | 90.26 |
| ChunkySeal | 37.12 | 88.97 | 92.74 | 96.05 | 98.35 | 99.63 | 37.12 | 99.91 | 99.63 | 99.08 | 98.53 | 96.88 | 37.12 | 100 | 100 | 99.91 | 99.82 | 99.63 |
| Ours | 45.45 | 99.41 | 99.61 | 99.71 | 99.91 | 100 | 40.17 | 100 | 100 | 100 | 99.80 | 98.73 | 43.07 | 100 | 100 | 99.90 | 99.32 | 98.75 |

| Method | Dropout PSNR(dB) | 0.30 | 0.40 | 0.50 | 0.60 | 0.70 | Gaussian Blur PSNR(dB) | 2 | 4 | 6 | 8 | 10 | Median Filter PSNR(dB) | 7 | 9 | 11 | 13 | 15 |
|---|---|---|---|---|---|---|---|---|---|---|---|---|---|---|---|---|---|---|
| HiDDeN | 33.60 | 59.19 | 62.04 | 65.90 | 69.30 | 71.60 | 33.60 | 53.03 | 52.48 | 52.76 | 52.85 | 53.03 | 33.60 | 51.10 | 50.09 | 50.55 | 49.36 | 49.72 |
| LISO | 33.04 | 88.19 | 95.63 | 98.30 | 99.54 | 99.82 | 33.04 | 67.83 | 56.71 | 56.80 | 56.76 | 56.62 | 33.04 | 62.04 | 50.69 | 51.70 | 51.47 | 51.47 |
| SSL | 35.12 | 61.21 | 67.46 | 73.16 | 76.75 | 79.14 | 35.12 | 70.22 | 63.97 | 62.50 | 61.67 | 61.67 | 35.12 | 68.57 | 52.21 | 51.01 | 51.19 | 51.47 |
| FIN | 35.74 | 98.05 | 99.32 | 99.61 | 99.90 | 100 | 35.74 | 95.70 | 96.68 | 96.58 | 96.58 | 96.48 | 35.74 | 92.87 | 60.35 | 88.18 | 43.75 | 76.46 |
| MBRS | 35.21 | 96.78 | 99.26 | 99.72 | 100 | 100 | 35.21 | 90.72 | 89.34 | 89.52 | 89.52 | 89.61 | 35.21 | 93.57 | 90.53 | 87.87 | 82.54 | 77.48 |
| CIN | 36.18 | 93.11 | 98.07 | 99.63 | 99.91 | 100 | 36.18 | 99.17 | 99.17 | 99.08 | 98.81 | 98.81 | 36.18 | 98.90 | 96.05 | 92.56 | 85.20 | 81.89 |
| LM | 36.28 | 100 | 100 | 100 | 100 | 100 | 36.28 | 100 | 100 | 100 | 100 | 100 | 36.28 | 99.90 | 6.45 | 49.12 | 91.60 | 17.58 |
| VideoSeal | 38.82 | 80.51 | 89.06 | 95.59 | 99.36 | 99.54 | 38.82 | 98.53 | 96.14 | 95.31 | 94.85 | 94.67 | 38.82 | 98.07 | 86.12 | 66.08 | 49.91 | 47.33 |
| ChunkySeal | 37.12 | 94.49 | 97.24 | 99.36 | 99.72 | 100 | 37.12 | 99.91 | 99.72 | 99.72 | 99.72 | 99.63 | 37.12 | 99.82 | 98.99 | 97.52 | 88.79 | 74.17 |
| Ours | 51.91 | 100 | 100 | 100 | 100 | 100 | 49.42 | 100 | 100 | 100 | 100 | 100 | 50.09 | 100 | 100 | 100 | 100 | 100 |

| Method | Shear PSNR(dB) | 35 | 45 | 55 | 65 | 75 | Rotate PSNR(dB) | 15 | 45 | 75 | 105 | 135 | Elastic PSNR(dB) | 1.0 | 1.5 | 2.0 | 2.5 | 3.0 |
|---|---|---|---|---|---|---|---|---|---|---|---|---|---|---|---|---|---|---|
| HiDDeN | 33.60 | 65.58 | 59.01 | 58.27 | 52.62 | 51.42 | 33.60 | 65.81 | 55.97 | 50.46 | 50.83 | 48.99 | 33.60 | 72.98 | 70.59 | 69.85 | 69.85 | 67.00 |
| LISO | 33.04 | 47.79 | 51.22 | 48.67 | 49.20 | 48.37 | 33.04 | 54.50 | 49.93 | 50.48 | 50.30 | 49.47 | 33.04 | 76.15 | 69.16 | 65.21 | 61.44 | 62.41 |
| SSL | 35.12 | 58.69 | 54.04 | 55.74 | 50.09 | 50.64 | 35.12 | 62.13 | 68.11 | 59.79 | 55.70 | 56.07 | 35.12 | 72.89 | 68.11 | 64.52 | 61.95 | 59.93 |
| FIN | 35.74 | 65.82 | 60.55 | 69.43 | 57.37 | 54.15 | 35.74 | 55.91 | 50.05 | 51.22 | 48.97 | 51.12 | 35.74 | 82.81 | 82.13 | 78.61 | 75.00 | 70.90 |
| MBRS | 35.21 | 58.13 | 56.57 | 56.07 | 52.76 | 54.96 | 35.21 | 59.05 | 51.06 | 48.07 | 51.01 | 50.92 | 35.21 | 97.33 | 97.33 | 97.24 | 96.05 | 92.83 |
| CIN | 36.18 | 99.68 | 99.63 | 99.72 | 80.15 | 62.64 | 36.18 | 91.31 | 98.90 | 58.36 | 55.88 | 55.15 | 36.18 | 99.36 | 98.99 | 98.16 | 95.86 | 93.84 |
| LM | 36.28 | 46.73 | 50.24 | 60.30 | 50.54 | 47.71 | 36.28 | 50.93 | 49.76 | 49.90 | 49.51 | 47.61 | 36.28 | 51.95 | 50.49 | 51.46 | 52.34 | 52.73 |
| VideoSeal | 38.82 | 99.82 | 99.72 | 99.17 | 77.67 | 55.97 | 38.82 | 99.82 | 99.45 | 50.69 | 51.38 | 48.81 | 38.82 | 99.82 | 99.63 | 99.36 | 99.08 | 97.70 |
| ChunkySeal | 37.12 | 98.94 | 96.51 | 86.81 | 51.75 | 48.99 | 37.12 | 99.59 | 96.42 | 51.79 | 49.13 | 50.41 | 37.12 | 100 | 99.91 | 99.82 | 100 | 99.36 |
| Ours | 46.13 | 100 | 99.90 | 99.95 | 99.38 | 99.83 | 49.39 | 100 | 100 | 100 | 100 | 100 | 44.13 | 100 | 100 | 99.80 | 99.71 | 98.04 |

| Method | Erase PSNR(dB) | 0.3 | 0.4 | 0.5 | 0.6 | 0.7 | Crop&Resize PSNR(dB) | 0.3 | 0.4 | 0.5 | 0.6 | 0.7 | Hue PSNR(dB) | -0.2 | -0.1 | 0.0 | +0.1 | +0.2 |
|---|---|---|---|---|---|---|---|---|---|---|---|---|---|---|---|---|---|---|
| HiDDeN | 33.60 | 76.56 | 74.54 | 70.59 | 67.92 | 61.58 | 33.60 | 55.42 | 58.09 | 63.42 | 66.64 | 69.67 | 33.60 | 57.63 | 67.83 | 78.58 | 69.76 | 59.01 |
| LISO | 33.04 | 99.17 | 99.08 | 97.98 | 96.97 | 96.14 | 33.04 | 50.00 | 54.46 | 50.92 | 50.97 | 52.44 | 33.04 | 65.72 | 95.54 | 100 | 98.07 | 74.63 |
| SSL | 35.12 | 60.29 | 56.07 | 54.60 | 54.50 | 51.56 | 35.12 | 55.24 | 57.72 | 59.19 | 62.04 | 63.69 | 35.12 | 74.26 | 79.32 | 82.81 | 79.23 | 73.44 |
| FIN | 35.74 | 96.09 | 94.53 | 89.06 | 86.72 | 81.54 | 35.74 | 52.05 | 48.73 | 52.73 | 53.61 | 57.42 | 35.74 | 100 | 100 | 100 | 100 | 100 |
| MBRS | 35.21 | 97.24 | 96.60 | 93.20 | 90.44 | 88.60 | 35.21 | 50.83 | 55.06 | 53.03 | 52.94 | 64.89 | 35.21 | 86.67 | 99.82 | 100 | 99.45 | 90.44 |
| CIN | 36.18 | 100 | 99.82 | 99.72 | 99.54 | 99.08 | 36.18 | 56.99 | 69.67 | 85.48 | 83.36 | 65.07 | 36.18 | 85.39 | 99.82 | 100 | 99.91 | 85.39 |
| LM | 36.28 | 90.92 | 88.38 | 83.11 | 79.39 | 76.37 | 36.28 | 48.93 | 50.20 | 47.66 | 49.32 | 50.98 | 36.28 | 94.43 | 100 | 100 | 100 | 95.12 |
| VideoSeal | 38.82 | 100 | 99.82 | 99.45 | 98.35 | 96.42 | 38.82 | 50.28 | 60.28 | 98.16 | 85.48 | 64.43 | 38.82 | 100 | 100 | 100 | 100 | 99.91 |
| ChunkySeal | 37.12 | 99.82 | 99.63 | 99.45 | 98.71 | 97.70 | 37.12 | 65.44 | 88.60 | 98.81 | 98.71 | 95.86 | 37.12 | 81.34 | 99.91 | 100 | 99.91 | 95.96 |
| Ours | 45.59 | 100 | 100 | 100 | 99.51 | 99.02 | 40.44 | 98.01 | 99.41 | 99.32 | 99.61 | 99.71 | 50.11 | 100 | 100 | 100 | 100 | 100 |

| Method | Brightness PSNR(dB) | 0.1 | 0.2 | 1.0 | 2.0 | 2.5 | Contrast PSNR(dB) | 0.1 | 0.2 | 1.0 | 2.0 | 2.5 | Saturation PSNR(dB) | 0.1 | 0.2 | 1.0 | 2.0 | 2.5 |
|---|---|---|---|---|---|---|---|---|---|---|---|---|---|---|---|---|---|---|
| HiDDeN | 33.60 | 55.24 | 63.97 | 78.49 | 64.25 | 60.57 | 33.60 | 58.55 | 66.45 | 72.98 | 71.78 | 67.28 | 33.60 | 62.41 | 69.03 | 73.45 | 75.09 | 74.45 |
| LISO | 33.04 | 98.76 | 99.54 | 100 | 93.43 | 90.07 | 33.04 | 99.54 | 99.82 | 100 | 99.54 | 98.76 | 33.04 | 94.30 | 98.81 | 100 | 99.82 | 99.68 |
| SSL | 35.12 | 68.93 | 77.48 | 83.31 | 69.67 | 60.29 | 35.12 | 69.67 | 79.32 | 82.63 | 77.39 | 70.96 | 35.12 | 71.60 | 77.67 | 83.36 | 78.68 | 75.37 |
| FIN | 35.74 | 88.18 | 98.54 | 100 | 89.75 | 88.09 | 35.74 | 84.18 | 97.66 | 100 | 99.80 | 99.41 | 35.74 | 100 | 100 | 100 | 100 | 99.90 |
| MBRS | 35.21 | 84.19 | 90.90 | 100 | 92.56 | 89.06 | 35.21 | 81.07 | 89.98 | 100 | 99.82 | 99.45 | 35.21 | 81.53 | 90.62 | 100 | 100 | 100 |
| CIN | 36.18 | 97.33 | 99.17 | 99.90 | 96.14 | 93.93 | 36.18 | 97.15 | 99.17 | 100 | 100 | 97.79 | 36.18 | 97.79 | 99.17 | 100 | 100 | 100 |
| LM | 36.28 | 100 | 100 | 100 | 90.72 | 86.91 | 36.28 | 100 | 100 | 100 | 99.90 | 99.80 | 36.28 | 100 | 100 | 100 | 100 | 100 |
| VideoSeal | 38.82 | 97.61 | 100 | 100 | 93.29 | 90.35 | 38.82 | 95.86 | 99.91 | 100 | 99.63 | 99.17 | 38.82 | 100 | 100 | 100 | 100 | 99.91 |
| ChunkySeal | 37.12 | 99.26 | 99.54 | 99.98 | 93.29 | 88.60 | 37.12 | 98.99 | 99.54 | 100 | 99.63 | 98.62 | 37.12 | 99.63 | 99.91 | 100 | 100 | 100 |
| Ours | 46.28 | 100 | 100 | 100 | 100 | 98.14 | 48.46 | 100 | 100 | 100 | 100 | 100 | 51.78 | 100 | 100 | 100 | 100 | 100 |

*Table 13.* Single-noise fine-tuning evaluation for the baseline methods. Baselines are fine-tuned for 10 epochs under each specific distortion; we report the stego PSNR (dB) and decoding accuracy (%) under different strengths of a single distortion.

| Method | JPEG | | | | | | Gaussian Noise | | | | | | S&P | | | | | |
|---|---|---|---|---|---|---|---|---|---|---|---|---|---|---|---|---|---|---|
| | PSNR(dB) | 50 | 60 | 70 | 80 | 90 | PSNR(dB) | 0.03 | 0.04 | 0.05 | 0.06 | 0.07 | PSNR(dB) | 0.02 | 0.05 | 0.10 | 0.15 | 0.20 |
| HiDDeN-ft | 35.11 | 55.56 | 55.38 | 56.86 | 58.85 | 57.03 | 34.26 | 65.62 | 67.53 | 67.19 | 67.36 | 66.93 | 34.31 | 74.23 | 68.84 | 67.97 | 65.89 | 61.89 |
| FIN-ft | 42.37 | 87.93 | 94.01 | 97.66 | 99.48 | 99.65 | 39.47 | 100 | 99.91 | 99.91 | 99.91 | 99.74 | 38.68 | 99.83 | 99.74 | 98.26 | 93.23 | 86.11 |
| MBRS-ft | 39.17 | 94.70 | 96.09 | 97.48 | 98.09 | 98.70 | 41.58 | 99.39 | 99.31 | 99.13 | 99.22 | 99.39 | 43.82 | 100 | 100 | 100 | 100 | 99.91 |
| CIN-ft | 40.01 | 95.49 | 96.01 | 97.74 | 98.35 | 98.70 | 39.24 | 100 | 100 | 100 | 100 | 99.91 | 39.63 | 100 | 100 | 100 | 100 | 100 |
| LM-ft | 41.87 | 99.57 | 100 | 99.83 | 100 | 100 | 41.08 | 100 | 100 | 100 | 100 | 100 | 41.17 | 100 | 100 | 100 | 100 | 99.65 |
| VideoSeal-ft | 41.18 | 96.69 | 97.98 | 98.90 | 99.91 | 100 | 37.90 | 100 | 98.99 | 99.36 | 98.25 | 96.51 | 39.40 | 100 | 100 | 99.54 | 98.25 | 94.30 |
| ChunkySeal-ft | 40.42 | 83.64 | 89.80 | 93.01 | 96.88 | 98.99 | 40.41 | 99.36 | 99.17 | 97.79 | 96.88 | 96.05 | 40.49 | 100 | 99.91 | 99.72 | 99.45 | 99.45 |
| Ours | 45.45 | 99.41 | 99.61 | 99.71 | 99.91 | 100 | 40.17 | 100 | 100 | 100 | 99.80 | 98.73 | 43.07 | 100 | 100 | 99.90 | 99.32 | 98.75 |

| Method | Dropout | | | | | | Gaussian Blur | | | | | | Median Filter | | | | | |
|---|---|---|---|---|---|---|---|---|---|---|---|---|---|---|---|---|---|---|
| | PSNR(dB) | 0.30 | 0.40 | 0.50 | 0.60 | 0.70 | PSNR(dB) | 2 | 4 | 6 | 8 | 10 | PSNR(dB) | 7 | 9 | 11 | 13 | 15 |
| HiDDeN-ft | 34.37 | 60.42 | 63.37 | 66.28 | 69.97 | 73.45 | 34.29 | 59.11 | 63.54 | 63.80 | 63.72 | 63.89 | 34.25 | 57.81 | 54.95 | 54.60 | 55.12 | 52.69 |
| FIN-ft | 47.70 | 94.18 | 96.61 | 98.44 | 99.05 | 99.48 | 46.04 | 95.66 | 99.39 | 99.39 | 99.39 | 99.57 | 45.14 | 79.25 | 71.70 | 73.00 | 63.37 | 66.67 |
| MBRS-ft | 47.97 | 98.18 | 99.22 | 99.74 | 99.83 | 100 | 47.22 | 89.41 | 100 | 100 | 100 | 100 | 42.62 | 97.31 | 97.14 | 96.27 | 95.31 | 92.88 |
| CIN-ft | 47.09 | 99.83 | 100 | 100 | 100 | 100 | 47.11 | 99.39 | 100 | 100 | 100 | 100 | 45.28 | 94.10 | 92.88 | 91.06 | 88.54 | 85.07 |
| LM-ft | 50.07 | 98.87 | 99.83 | 99.74 | 99.91 | 99.91 | 48.11 | 99.91 | 100 | 100 | 100 | 100 | 38.86 | 99.65 | 12.59 | 54.17 | 73.26 | 36.55 |
| VideoSeal-ft | 44.93 | 84.56 | 92.19 | 95.77 | 98.35 | 98.99 | 42.55 | 96.14 | 95.31 | 95.13 | 94.39 | 94.30 | 44.42 | 90.07 | 84.10 | 80.33 | 76.75 | 67.65 |
| ChunkySeal-ft | 47.64 | 96.78 | 98.81 | 99.45 | 99.82 | 99.91 | 47.45 | 97.61 | 97.43 | 97.33 | 97.15 | 96.97 | 47.37 | 93.66 | 92.00 | 90.44 | 86.95 | 80.33 |
| Ours | 51.91 | 100 | 100 | 100 | 100 | 100 | 49.42 | 100 | 100 | 100 | 100 | 100 | 50.09 | 100 | 100 | 100 | 100 | 100 |

| Method | Shear | | | | | | Rotate | | | | | | Elastic | | | | | |
|---|---|---|---|---|---|---|---|---|---|---|---|---|---|---|---|---|---|---|
| | PSNR(dB) | 35 | 45 | 55 | 65 | 75 | PSNR(dB) | 15 | 45 | 75 | 105 | 135 | PSNR(dB) | 1.0 | 1.5 | 2.0 | 2.5 | 3.0 |
| HiDDeN-ft | 34.14 | 64.32 | 61.85 | 61.41 | 55.38 | 52.69 | 34.14 | 61.11 | 54.69 | 51.87 | 49.91 | 50.09 | 34.23 | 70.31 | 68.92 | 69.18 | 67.80 | 67.97 |
| FIN-ft | 41.78 | 70.31 | 66.28 | 67.01 | 61.15 | 56.99 | 44.79 | 59.20 | 49.87 | 50.65 | 50.43 | 50.52 | 40.34 | 95.57 | 93.75 | 92.01 | 86.63 | 79.77 |
| MBRS-ft | 42.05 | 76.48 | 89.58 | 89.54 | 64.89 | 55.38 | 37.78 | 69.49 | 65.97 | 65.03 | 55.30 | 52.91 | 44.03 | 99.83 | 99.57 | 99.39 | 98.18 | 96.27 |
| CIN-ft | 41.86 | 99.91 | 99.96 | 99.91 | 99.39 | 89.50 | 44.96 | 87.80 | 91.58 | 89.15 | 90.23 | 90.41 | 39.70 | 99.83 | 99.83 | 98.96 | 97.74 | 96.88 |
| LM-ft | 39.14 | 52.17 | 54.56 | 59.64 | 52.95 | 49.70 | 42.77 | 52.73 | 52.60 | 50.00 | 49.83 | 50.30 | 35.65 | 98.18 | 98.00 | 97.05 | 96.18 | 92.88 |
| VideoSeal-ft | 39.46 | 99.82 | 99.54 | 98.76 | 87.04 | 60.25 | 46.44 | 93.61 | 90.95 | 50.41 | 50.97 | 49.86 | 40.71 | 100 | 99.91 | 99.82 | 99.45 | 98.16 |
| ChunkySeal-ft | 40.45 | 99.17 | 97.98 | 95.08 | 81.34 | 58.00 | 47.43 | 94.03 | 90.21 | 52.76 | 51.15 | 50.60 | 40.48 | 99.82 | 99.45 | 99.91 | 99.63 | 98.62 |
| Ours | 46.13 | 100 | 99.90 | 99.95 | 99.38 | 99.83 | 49.39 | 100 | 100 | 100 | 100 | 100 | 44.13 | 100 | 100 | 99.80 | 99.71 | 98.04 |

| Method | Erase | | | | | | Crop&Resize | | | | | | Hue | | | | | |
|---|---|---|---|---|---|---|---|---|---|---|---|---|---|---|---|---|---|---|
| | PSNR(dB) | 0.3 | 0.4 | 0.5 | 0.6 | 0.7 | PSNR(dB) | 0.3 | 0.4 | 0.5 | 0.6 | 0.7 | PSNR(dB) | -0.2 | -0.1 | 0.0 | +0.1 | +0.2 |
| HiDDeN-ft | 34.17 | 69.27 | 68.23 | 66.32 | 65.10 | 60.85 | 33.86 | 60.24 | 65.89 | 66.67 | 66.93 | 65.28 | 34.26 | 56.77 | 67.01 | 72.22 | 67.45 | 60.07 |
| FIN-ft | 40.06 | 99.31 | 97.83 | 94.36 | 91.41 | 89.93 | 36.05 | 52.00 | 58.25 | 58.59 | 63.63 | 66.93 | 45.99 | 99.91 | 99.91 | 100 | 100 | 99.91 |
| MBRS-ft | 45.02 | 99.13 | 98.87 | 97.66 | 97.57 | 96.44 | 35.40 | 53.65 | 82.47 | 100 | 96.09 | 84.64 | 47.23 | 96.18 | 100 | 100 | 100 | 97.57 |
| CIN-ft | 42.04 | 100 | 100 | 99.91 | 99.41 | 98.93 | 36.71 | 80.41 | 90.41 | 95.24 | 98.75 | 99.12 | 46.09 | 99.22 | 100 | 100 | 100 | 99.65 |
| LM-ft | 42.95 | 90.89 | 86.72 | 82.81 | 77.26 | 73.00 | 34.65 | 53.39 | 52.95 | 57.47 | 58.51 | 61.11 | 47.32 | 89.58 | 99.91 | 100 | 100 | 91.67 |
| VideoSeal-ft | 41.20 | 100 | 99.91 | 99.36 | 99.08 | 98.62 | 38.33 | 97.92 | 98.74 | 99.11 | 99.38 | 99.63 | 45.59 | 98.71 | 99.36 | 99.82 | 98.99 | 99.08 |
| ChunkySeal-ft | 40.64 | 99.82 | 99.45 | 99.36 | 97.98 | 97.06 | 33.44 | 92.28 | 98.16 | 99.82 | 100 | 99.82 | 47.49 | 87.68 | 99.36 | 99.63 | 99.45 | 93.84 |
| Ours | 45.59 | 100 | 100 | 100 | 99.51 | 99.02 | 40.44 | 98.01 | 99.41 | 99.32 | 99.61 | 99.71 | 50.11 | 100 | 100 | 100 | 100 | 100 |

| Method | Brightness | | | | | | Contrast | | | | | | Saturation | | | | | |
|---|---|---|---|---|---|---|---|---|---|---|---|---|---|---|---|---|---|---|
| | PSNR(dB) | 0.1 | 0.2 | 1.0 | 2.0 | 2.5 | PSNR(dB) | 0.1 | 0.2 | 1.0 | 2.0 | 2.5 | PSNR(dB) | 0.1 | 0.2 | 1.0 | 2.0 | 2.5 |
| HiDDeN-ft | 34.26 | 58.07 | 61.81 | 71.27 | 61.72 | 59.46 | 34.74 | 57.64 | 64.50 | 73.09 | 68.06 | 65.19 | 34.91 | 61.20 | 64.50 | 73.35 | 70.31 | 69.18 |
| FIN-ft | 39.33 | 100 | 100 | 100 | 94.97 | 91.75 | 44.22 | 99.65 | 99.74 | 100 | 98.35 | 97.92 | 45.83 | 99.91 | 99.91 | 100 | 99.91 | 99.31 |
| MBRS-ft | 40.73 | 100 | 100 | 100 | 95.57 | 92.36 | 45.56 | 99.74 | 100 | 100 | 99.13 | 97.57 | 47.23 | 98.61 | 99.22 | 100 | 100 | 99.83 |
| CIN-ft | 39.61 | 100 | 100 | 100 | 97.57 | 96.09 | 43.59 | 100 | 100 | 100 | 100 | 99.91 | 46.23 | 100 | 100 | 100 | 100 | 100 |
| LM-ft | 40.78 | 100 | 100 | 100 | 91.67 | 87.59 | 45.60 | 100 | 100 | 100 | 98.44 | 97.57 | 48.16 | 100 | 100 | 99.31 | 100 | 99.74 |
| VideoSeal-ft | 40.74 | 99.91 | 99.91 | 100 | 94.30 | 90.62 | 44.93 | 98.53 | 99.08 | 99.72 | 97.52 | 96.51 | 45.99 | 99.54 | 99.45 | 99.63 | 99.17 | 98.81 |
| ChunkySeal-ft | 40.49 | 99.82 | 100 | 100 | 95.31 | 91.18 | 44.16 | 99.72 | 100 | 100 | 99.45 | 98.35 | 47.57 | 99.26 | 99.72 | 99.72 | 99.36 | 99.08 |
| Ours | 46.28 | 100 | 100 | 100 | 100 | 98.14 | 48.46 | 100 | 100 | 100 | 100 | 100 | 51.78 | 100 | 100 | 100 | 100 | 100 |

