# OpenReview forum: "Revisiting Coding-Based Approaches to Overcome the Curse of Dimensionality in Learning-Based Watermarking"
_ICML.cc/2026/Conference — ICML 2026 regular_

### Official Review · Reviewer_oV5E · 2026-03-03

**Soundness:** 1
**Presentation:** 3
**Significance:** 3
**Originality:** 3
**Overall Recommendation:** 4
**Confidence:** 5

**Summary:**

This submission proposes combining a side-informed modulation scheme (ST-QIM) from the early days of watermarking with learning-based feature extraction from the new deep-watermarking era. The result is an original watermarking scheme for still images.

**Compliance With Llm Reviewing Policy:**

Affirmed.

**Final Justification:**

Great discussion with the authors that improves the quality of the paper.

**Key Questions For Authors:**

Minor comments:
- Fig. 1. ChunkySeal should be a single point in the graph, not a plot.

Major comments:
- The control of the embedding distortion is not detailed. Eq. (19) suggests that it relies on the hyperparameters $\lambda_1$ and  $\lambda_2$, or the options in the ablation studies.... I suppose it also depends on $\Delta$ in (14). Moreover, Fig. 1 suggests that the embedding distortion can be accurately tuned to a given target PSNR (here 50dB). How is this possible?

- I am interested in an ablation study about the role of the DWT.

- Table 4 is confusing, especially with the comment:
> illustrating the extent to which inference stage adjustment can shift the operating point
between visual quality and robustness without retraining

Does this mean that, for any attack in Table 4, the embedding is tuned to face that attack and only that one? ie. the noise layer used at inference time is this unique attack. How do you compute the gradient (20) at inference time if this noise layer is not differentiable (eg. JPEG)? Is it fair to compare the robustness against one specific attack when the competitors were trained to gain robustness against many attacks, and the proposed scheme was trained only against this unique attack?

**Limitations:**

The authors did not discuss a strong limitation: the embedding runtime! The watermark embedding is obtained via gradient descent, requiring 3000 iterations.

**Strengths And Weaknesses:**

## Strengths
- S1: The submission is well written, and the maths are ok.
- S2: I appreciate the bibliography citing numerous studies from the early days of watermarking (which most recent papers tend to forget).
- S3: The contribution is significant as it proposes a hybrid approach with a sound modulation scheme compared to full fledged deep learning based watermarking scheme. Also, this scheme offers some flexibility in the trade-off between robustness and capacity

## Weaknesses
- W1: This approach is not new. [A], in 2020, combines a zero-bit side-informed modulation scheme with a deep learning feature extraction. [B] combines the TrunCated Cyclic Shift Keying with a deep learning feature extraction. And of course, SSL, cited in the submission, follows exactly the same line. Therefore, statements like
> We propose a framework that decouples robust feature extraction from message encoding.

are misleading in the sense that this is not at all a novelty.

- W2: The presentation should be clearer w.r.t. to the learning phase. As far as I understand, the loss (19) is used twice: once for training and once at inference time. Yet, the first phrase optimizes the feature extraction (ie. the model), the second the watermarked image (20). An equation detailing the first phase is expected, especially because i) it seems to be a double optimization w.r.t. the model parameters and the images, ii) the augmentation layer (aka noise layer) contains multiple transformations (is it an Expectation over Transformations?), some of them are not differentiable. In brief, many technical details are missing. Same questions pop up at inference time: there are T iterations; does the noise layer change from one iteration to the next?

- W3: ST-QIM is known to be suboptimal [C]. Why not directly use better side-informed modulation schemes (like DC-QIM with nested lattices, or treillis coding modulation). See [D]. This comment also ruins the introduction of the problem in section 3, which assumes, right from the beginning in (1), a bit-wise modulation with carriers $(u_k)_k$ (ie. 1 carrier = 1 bit). Section 3 is nothing more than the traditional development of the Spread Spectrum modulation scheme, which **obviously** needs orthogonal carriers to kill symbol interference. Note that, at this stage, the host interference in $y_i'$ (6) may have a bigger impact than the symbol interference. In other words, Section 3 hides a problem that is silently solved in Sect. 4 by (14) (ST-QIM aka SCS - Scalar Costa Scheme). This makes the point in Section 3 not so relevant. Indeed, SSL also uses orthogonal carriers, yet, in Fig.1, its accuracy drops with the payload: This shows that this is not just a problem of '*coding cross-talk*'. The provided interpretation is wrong.

[A] Are Classification Deep Neural Networks Good for Blind Image Watermarking? https://www.mdpi.com/1099-4300/22/2/198

[B] SWIFT: Semantic Watermarking for Image Forgery Thwarting. https://arxiv.org/pdf/2407.18995

[C] Spread-Spectrum vs. Quantization-Based Data Hiding: Misconceptions and Implications. Perez-Freire &Perez-Gonzalez. https://www.researchgate.net/profile/Luis-Perez-Freire/publication/221011221_Spread-spectrum_vs_quantization-based_data_hiding_Misconceptions_and_implications/links/5456a9340cf2bccc490f2b4e/Spread-spectrum-vs-quantization-based-data-hiding-Misconceptions-and-implications.pdf

[D]  Data-hiding codes. Moulin & Koetter. https://ieeexplore.ieee.org/stamp/stamp.jsp?arnumber=1545764

- W4: The experimental part is not convincing for the following reasons
  - The ablation study w.r.t. orthogonal vs. correlated carriers (Sect. 5.4) brings **obvious** results for a watermarking expert. Note that "correlated projection matrix" is not defined, therefore not reproducible.
  - The ablation study "QIM-style loss" vs."MSE loss" is also a mystery. The proposal uses the QIM-style loss defined by (16), (17), and (18), which are all based on MSE. Therefore, I cannot imagine what "MSE loss" means.
  - Images of size 128 x 128 are not realistic. They are smaller than thumbnails on the Internet; website thumbnails are 250x250. In real-life, today's images contain millions of pixels. It raises the question of whether the proposed scheme can handle larger images, and images of different sizes.
   - I have major doubts about the reported accuracy results against Shear and Elastic transformations. I am sure that Hidden, VideoSeal, and SSL are NOT robust to these super-strong attacks at all. I am very surprised that their reported accuracy is not 50%.
Moreover, the proposed scheme is based on INN, which are known to be weak against geometric transformations, and the DWT used upfront is likewise. I am very surprised by the robustness against geometric attacks knowing that the backbone is DWT+INN.

---

> ### Author Rebuttal · Authors · 2026-03-31
>
> **Results:  https://anonymous.4open.science/r/Rebuttal-ICML-9170-BB112010/Rebuttal_ICML2026_9170.pdf**
>
> ---
>
> **W1:** Our novelty was never meant to rest on combining deep features with traditional coding alone. What distinguishes OrthoMark: **(1)** The SIR framework is the first theoretical diagnosis of high-capacity collapse, proving per-bit SIR degrades rapidly with payload until accuracy collapses toward chance. None of [A], [B], or SSL address this. **(2)** Every prior method collapses to ~50% at high payloads even on clean images (Figure 1); OrthoMark first achieves reliable decoding across 15 distortions. **(3)** Dimensions absent from all cited works: **(i)** root-cause analysis of high-capacity failure motivating the entire design; **(ii)** End-to-end joint training of the DL model and the coding objective, which Table 5 shows is structurally necessary.
>
> ---
>
> **W2:** There is just **one optimization phase**: RFE parameters and watermarked image x are updated **simultaneously** via single backpropagation through loss (19); x itself plays the encoder's role. The noise layer does **not** use EoT: we **randomly sample one distortion per step** (same as MBRS/FIN/CIN), implemented via **Kornia** which provides differentiable implementations of various noises; non-differentiable versions used at test time. At inference, RFE is frozen; x is optimized for T steps with the same loss. Our code will be open-sourced later.
>
> ---
>
> **W3:** DC-QIM achieves similar performance to ST-QIM (Table 11, linked). Our framework is agnostic to the specific QIM variant; any suitable variant can be adopted.
>
> Section 3 addresses a gap that, despite the maturity of classical spread-spectrum theory, had not been bridged in the deep watermarking literature. Recently, **ChunkySeal (2025) scaled up VideoSeal by several orders of magnitude, but only achieved a 4x payload gain.** Section 3 contributes two insights absent from classical analysis: **(1)** classical theory assumes orthogonal carriers given; we imply that learned carriers lack this guarantee will cause inevitable SIR degradation; **(2)** classical carriers are predefined/shared with perfect alignment; deep encoder-decoder pairs cannot guarantee this. These insights motivate our design: a predefined orthogonal space and elimination of the encoder in favor of directly optimizing the cover image. Without this formulation, these choices cannot be justified.
>
> **Host interference.** Assumption 3.1 isolates crosstalk deliberately, because host interference is not the root cause: it can be suppressed by increasing embedding strength, whereas crosstalk from correlated carriers cannot. If host interference dominated, prior methods would succeed at high capacity under unconstrained strength; they do not. Oth+MSE achieves full accuracy by increasing strength alone without host interference suppression, confirming it is not the capacity bottleneck.
>
> **SSL.** Its failure confirms our theory: **(i)** non-bijective ResNet cannot guarantee carrier non-collision at high payload (see Reviewer JJDj, Q2); **(ii)** its features, pretrained for classification rather than coding-based watermarking, are insufficient for high-capacity decoding.
>
> ---
>
> **W4:**
>
> **Correlated Carrier** is V = ρ·u + (1−ρ)·ε, where u is a shared direction and ε is an orthonormal basis. ρ controls inter-carrier correlation.
>
> **MSE vs. QIM loss.** Both use MSE but differ in targets. **MSE baseline**: L=MSE(t,m), m∈{±1}ᵐ, fixed scalar targets (as in MBRS/FIN/CIN). **QIM loss**: targets nearest correct QIM coset point, a dynamic structure-aware target.
>
> **Resolution.** Table 9 (linked): same checkpoint evaluated *without retraining* across various resolutions and payloads.
>
> **Geometric robustness.** We did not use official checkpoints, which lack geometric distortion training and yield 50% as mentioned. We retrained *every* baseline from scratch under our identical 15-distortion noise layer, requiring substantial effort. All checkpoints will be released then.
>
> ---
>
> **Q1:** We retrained it across 2⁶–2¹⁰ to provide a full capacity curve with Figure 1.
>
> **Q2:** PSNR is controlled in two steps: during training, the quality loss weight is tuned near 50dB; at inference, the residual is scaled by α=10^((PSNR_current−PSNR_target)/20), yielding exact control without retraining.
>
> **Q3:** Table 12 (linked) ablates DWT. Removing it (INN backbone) reduces AVG from 99.59% to 97.79%, notably on JPEG and Elastic.
>
> **Q4:** The comparison is fair: all models in Table 4, including ours, use the checkpoint trained from Table 1. Our method only varies inference noise composition; no retraining occurs. Table 4 demonstrates a unique capability of ours: one checkpoint can be steered toward specific robustness profiles at inference, a flexibility prior methods lack.
>
> ---
> **L:** We acknowledge this limitation. Due to space constraints, we respectfully refer the you to our response to Reviewer JJDj (Q3), where we discuss the latency in detail.

---

> > ### Author Rebuttal · Reviewer_oV5E · 2026-04-02
> >
> > ## Minor comments
> > - There is a misunderstanding. By DC-QIM, I mean quantization on Euclidean lattices more general than $\delta\mathbb{Z}^n$ used in this paper. As explained in [D], these side-informed embeddings based on non-trivial lattices are more powerful than independently quantizing the coordinates (a.k.a. Scalar Costa Scheme).
> > - By bigger images, I mean images with $\gtrsim$ 1M pixels. Nowadays, no camera captures or no commercial AI generates an image smaller than this.
> > - Related to reviewer JJDj's comments: news agencies are very picky on latency, especially for live events. Some even guarantee a latency of less than 3 seconds (from image capture to availability to download from their website) to their clients. The EU AI Act Art. 50 just requires the detection of AI-generated content. *Embedding rich forensic metadata* is not required.
> > - - I agree that previous papers were limited, but the simple derivations in Sect. 3 are just well-known folklore in the watermarking literature.
> >
> > ## Major comments
> > - I am still very surprised by the robustness against geometric transformations because neither DWT nor INN is invariant to these attacks. How is this robustness achieved if the backbone is not robust? Could you swear that you are not inverse transforming the attacked image before querying the decoder?
> > - Table 4: I understand that the model is a unique checkpoint. I acknowledge that this is a good point. But I also understand that inference-time optimization assumes a single attack. Therefore, it is an embedding specific to each attack. This is unfair.

---

> > > ### Author Response · Authors · 2026-04-07
> > >
> > > **New results in: https://anonymous.4open.science/r/Rebuttal-ICML-9170-BB112010/2_Rebuttal_ICML_2026_9170.pdf**
> > >
> > > ---
> > >
> > > **Minor Comments**
> > >
> > > 1.
> > > Thank you for your further clarification, and we agree that the ST-QIM used in our method is one instance within the broader family of QIM. We further implemented an additional variant using 2-D hexagonal-lattice A₂ quantization.
> > >
> > >  As shown in Table 11 (linked), its performance is very close to the baseline, suggesting the primary gain comes from orthogonal projection and end-to-end training rather than the specific lattice geometry.
> > >
> > > This also confirms that **other QIM variants can be incorporated into our framework seamlessly.** More broadly, we agree that going beyond scalar quantization is a meaningful direction, and we appreciate your suggestion in highlighting it.
> > >
> > > ---
> > >
> > >  2.
> > > Thank you for the helpful comment. We evaluated our method at 4096×4096 (16M pixels); as shown in Table 13 (linked), performance remains effective. Since our method has no explicit encoder, no architectural changes are needed to align image and message features across resolutions. The projection space V is training-free, so adapting to arbitrary sizes only needs to adjust V. Our method thus scales without retraining, including to the megapixel regime.
> > >
> > > ---
> > >
> > > 3.
> > > Thank you for the comment.
> > >
> > > **(1) Scenario clarification.** We agree that ultra-low-latency pipelines impose strict constraints. Our work targets a complementary and equally important class of applications where robustness and capacity take priority over encoding speed, such as protecting high-value content like commercial posters and copyrighted artwork.
> > >
> > > **(2) The trade-off we target is practically meaningful.** Although our entire pipeline takes approximately 1 second, it yields substantial gains in both robustness and capacity. While not suited for real-time streaming, for offline workflows such as forensic verification and post-hoc tracing of high-value images, this overhead is worth the benefit.
> > >
> > > **(3) Emerging regulations go beyond binary detection.** While the EU AI Act Art. 50 focuses on detectability, the California AI Transparency Act (SB 942, amended by AB 853) already mentions embedding provider identity, creation time, and a unique identifier in AI-generated content, signaling that higher-capacity watermarking is a future regulatory trend. Moreover, **embedding richer metadata entails no drawback** and enhances traceability, opening broader prospects for practical deployment.
> > >
> > > ---
> > >
> > > 4.
> > > Thank you for the professional comments. We appreciate your deep expertise in the watermarking domain. We agree that, for experienced researchers in watermarking, the derivations in Sec. 3 may appear as well-known principles from the classical literature.
> > >
> > > However, we include this section for two reasons. First, many recent researchers, especially those entering the field through deep watermarking, may not be fully familiar with these classical insights. Providing a clear and self-contained formulation helps bridge this knowledge gap and makes the paper more accessible to a broader audience.
> > >
> > > Second, and more importantly, while these principles are well established in classical watermarking, they have not been properly understood or effectively realized in deep watermarking methods. A key contribution of our work is to revisit these classical insights, analyze why they do not naturally carry over to deep models, and demonstrate how they can be systematically integrated into modern deep watermarking pipelines. In this sense, the role of Sec. 3 is not merely to restate known results, but to establish the theoretical foundation that enables our subsequent method.
> > >
> > > Therefore, we present these derivations in this version.
> > >
> > > ---
> > >
> > > **Major comments**
> > >
> > > 1.
> > > We appreciate your thorough scrutiny. As ICML submitters, we take data integrity seriously and have provided full code in the supplement; we hope our discussion can proceed on mutual trust and good faith.
> > >
> > > To directly answer: no, we **do not apply inverse geometric transformation before decoding.** To address your concern: (1) We agree that DWT is not invariant to geometric distortions; however, through end-to-end training with INN, the overall pipeline learns geometrically robust representations. (2) We trained our method separately under geometric noises. As shown in Fig. 6 (linked), the learned embedding patterns adapt to each noise (e.g., ring-shaped for rotation, central-concentrated for shear, border-shifted for erasing), confirming that end-to-end training enables geometrically robust features. The INN-based CIN baseline (Table 14 linked) also exhibits geometric robustness, further corroborating this.
> > >
> > > 2.
> > > Thank you for recognizing our method's strength. Following your suggestion, we conducted a fairer and more practical evaluation: baselines from Table 1 were fine-tuned for 10 epochs under each specific noise. As shown in Table 14(linked), our method still retains a substantial advantage.

---

### Official Review · Reviewer_JJDj · 2026-03-07

**Soundness:** 3
**Presentation:** 2
**Significance:** 3
**Originality:** 2
**Overall Recommendation:** 4
**Confidence:** 4

**Summary:**

This paper investigates the "curse of dimensionality" problem in deep learning-based watermarking techniques, where decoding accuracy drops sharply under high-capacity payloads. To address this issue, the authors propose OrthoMark, an innovative framework that decouples robust feature extraction (RFE) from structured coding and decoding (SED). This method first utilizes a deep network to extract distortion-invariant features in the wavelet domain, then applies quantization index modulation (QIM) and orthogonal projection matrices in the latent space, thereby forcing the use of orthogonal bit carriers to suppress coded crosstalk. Experimental results demonstrate that OrthoMark successfully scales the payload capacity to an impressive 32,768 bits while maintaining high visual quality and robustness against various distortions.

**Compliance With Llm Reviewing Policy:**

Affirmed.

**Key Questions For Authors:**

1. The proposed model requires 3000 optimization iterations during the inference phase to embed the watermark. In contrast, most baseline models (e.g., MBRS and VideoSeal) require only one forward propagation. This results in a highly unfair baseline comparison. The paper should provide a comprehensive table comparing inference latency. Furthermore, it is suggested that an equivalent computational budget be introduced for the baseline models during testing. This could involve using the baseline decoder to guide image optimization via gradients. Under these identical conditions, will OrthoMark still maintain its absolute advantage?

2. The paper explicitly mentions the use of invertible neural networks (INNs) in Section 4.1. However, it also explicitly states that its invertibility is not utilized. If this invertibility has no practical use, the paper should explain why INNs remain the best choice for robust feature extraction. Such an explanation can be provided through theoretical analysis or rigorous ablation experiments. Why are standard optimization architectures like ResNet or Vision Transformer insufficient for this task?

3. Processing high-dimensional payloads like 32,768 bits requires computing orthogonal projection and QIM loss. Performing 3,000 backpropagations on a single image consumes significant amounts of GPU memory and time. How should this framework be adapted to modern real-time watermarking scenarios? These practical applications typically require high query throughput. Are there potential ways to transform this optimization-based embedding method into a single-pass feedforward encoder?

**Limitations:**

yes

**Strengths And Weaknesses:**

Soundness:  The theoretical analysis is executed exceptionally well. The authors introduce a linear superposition model to explain the capacity bottleneck from the perspective of signal-to-noise ratio (SIR). This rigorously demonstrates why existing deep watermarking networks are inevitably affected by coding crosstalk under high payloads. The approach of abstracting the underlying mathematical bottleneck from empirical phenomena is commendable. However, the experimental evaluation suffers from a seriously unfair benchmark. To embed the watermark, OrthoMark freezes its parameters and uses up to 3000 backpropagation iterations to optimize the input image. In contrast, most baseline methods (e.g., HiDDeN, MBRS, VideoSeal) are standard feedforward networks. They require only one forward propagation, typically taking only a few milliseconds. Comparing an extremely resource-intensive optimization-based method to an ultra-lightweight single-step method is essentially comparing apples to oranges. This overwhelming performance advantage lacks scientific persuasiveness without strict control over computational budgets or inference latency.

Presentation:  The paper is well-structured and clearly written. The re-examination of the QIM mechanism and the visualization in Figure 2 are intuitive and easy to understand. However, the narrative deliberately downplays the extremely high computational cost of the method. The abstract and introduction emphasize the ability to achieve perfect decoding accuracy with extremely high payloads. In stark contrast, the severe limitation of requiring 3000 optimization iterations per image during the inference phase is subtly concealed. This requirement can be considered the system's fatal weakness. However, it only appears at the end of Section 4.2 and in the ablation experiments. This "selective presentation" strategy severely reduces the paper's academic objectivity by top-tier conference standards.

Significance: This paper breaks through the capacity limit of deep watermarking technology, proving that classical coding theory can perfectly complement robust deep features. This points to a promising direction for hybrid architectures in future research. However, the excessively high embedding latency makes it unsuitable for practical industrial applications. Real-world generative AI scenarios require extremely lightweight watermarking embeddings with near-zero latency. Performing thousands of gradient updates on a single image is clearly impractical, making the current framework unsuitable for time-critical systems.

Originality: Embedding orthogonal projection and QIM into a deep latent space is a clever reconstruction of the existing paradigm. However, the choice of core architectural components seems too arbitrary and lacks theoretical justification. The authors use invertible neural network (INN) modules as feature extractors, but they explicitly state in Section 4.1 that they do not utilize their invertibility. Abandoning these defining bijective properties raises a crucial question: Why not adopt a more computationally efficient and expressive architecture, such as ResNet or Vision Transformer? This approach exposes a logical contradiction in the overall design philosophy.

---

> ### Author Rebuttal · Authors · 2026-03-31
>
> **We have included some results in:  https://anonymous.4open.science/r/Rebuttal-ICML-9170-BB112010/Rebuttal_ICML2026_9170.pdf**
>
> ---
>
> **Soundness & Q1:** Thank you for this constructive suggestion. In Appendix Table 5, we freeze the decoders of all Table 1 models and apply our SED procedure under an identical budget (T = 3,000). Even the best-upgraded baseline (CIN+SED, 92.40%) remains 7.2 pp below ours (99.59%).
>
> More importantly, prior models consistently fail to improve under SED; the strongest baselines actually *degrade*. This reveals a fundamental limitation: prior models were trained under objectives that never required features to support orthogonal QIM coset structure, and no amount of inference-time optimization can compensate. This equal-budget comparison demonstrates that the learned representations of prior methods are the root bottleneck, and that end-to-end joint training of the feature extractor and coding objective is the necessary solution.
>
> ---
>
> **Significance & Q3:** Thank you for this valuable suggestion. Full latency measurements are provided in Table 8 (linked). We first note that T = 600 already surpasses all baselines in Table 1 with a favorable efficiency/robustness trade-off. We therefore adopt T = 600 as a more practical setting for the latency comparison.
>
> We acknowledge that OrthoMark's encoding latency is higher than single-pass methods. However, among optimization-based methods it is the most efficient: 2.9× faster than SSL and 6.2× faster than LISO, with strictly stronger robustness. On the decoding side, our decoder runs in a single forward pass (0.9 ms/image), comparable to FIN (1.1 ms) and MBRS (1.0 ms).
>
> Important real-world scenarios exist where high-capacity robustness is the primary requirement, and trading encoding time for stronger protection is both acceptable and necessary.
>
> **(1) Media provenance certification.** A news agency certifying photographs for long-term redistribution needs sufficient payload for structured provenance metadata. Each photograph is signed once but verified many times. Robustness and capacity are the foremost requirements; encoding latency is not time-critical. OrthoMark's single-pass decoder (0.9 ms/image) precisely addresses the high-throughput verification side.
>
> **(2) Generative AI content governance.** Embedding rich forensic metadata for attribution and abuse investigation demands both large capacity and strong robustness. No prior deep watermarking method can jointly satisfy these requirements; OrthoMark is the first to make this possible. Moreover, our per-image encoding time **(~1s)** does not introduce a significant bottleneck relative to the latency of generative models themselves.
>
> We appreciate the suggestion of a single-pass feedforward encoder via knowledge distillation as a promising future direction. We emphasize, however, **that as the first work to theoretically analyze the root cause of failure in high-capacity deep watermarking and to propose a principled solution, the contributions of this paper are already substantial.**
>
> ---
>
> **Originality & Q2:** Thank you for this question. The choice of INN is not arbitrary. "Invertibility is not utilized" (Section 4.1) refers to not calling the reverse path f⁻¹(z) at inference; the **bijective property** itself is deeply exploited.
>
> **Non-collision guarantee.** Embedding m₁ ≠ m₂ into the same cover produces distinct QIM coset targets z_{m₁} ≠ z_{m₂}; for decoding to succeed, the watermarked images must also be distinct. Bijectivity guarantees this by construction. Non-bijective architectures (e.g., ResNet, ViT) may map distinct images to identical features, rendering the optimization non-unique. As payload grows and cosets become denser, collisions grow frequent until decoding fails. This is the fundamental mechanism behind the capacity wall of non-bijective architectures.
>
> Table 12 (linked) confirms this: replacing INN with ResNet yields substantially degraded performance. Table 8 shows this substitution offers no efficiency advantage.
>
> ---
>
> **Presentation:** Thank you for this feedback. We will revise the abstract and introduction to discuss encoding latency explicitly, and add a dedicated Limitations section. We wish to provide context: prior methods had identified the high-capacity limitation, but none diagnosed its root cause or proposed a solution. Our core contribution is the first theoretical framework explaining *why* deep watermarking fails at high payload, together with a working solution. **We believe solving a previously open problem necessarily precedes optimizing efficiency, and delivering a capability no prior method could achieve is a meaningful advance in its own right.**

---

> > ### Author Rebuttal · Reviewer_JJDj · 2026-04-01
> >
> > I appreciate the authors' efforts in conducting and presenting the supplementary experiments during the rebuttal phase. These additional empirical results effectively address my initial concerns and provide clear, satisfactory answers to the questions I raised. Given that the theoretical or empirical gaps have been successfully filled by these new additions, I am satisfied with the revisions.

---

> > > ### Author Response · Authors · 2026-04-04
> > >
> > > Thank you very much for taking the time to carefully review our rebuttal and the supplementary experiments. We are glad that the additional results have adequately addressed your concerns. We also appreciate your recognition of the theoretical and empirical improvements we have made. Your constructive feedback has been invaluable in strengthening our work.

---

### Official Review · Reviewer_ScsT · 2026-03-12

**Soundness:** 2
**Presentation:** 3
**Significance:** 4
**Originality:** 3
**Overall Recommendation:** 4
**Confidence:** 3

**Summary:**

The paper under review proposes OrthoMark. This is a framework that decouples
robust feature extraction from message encoding. It first learns a distortion-invariant feature representation using a deep robust feature extractor, and then performs watermark encoding
and decoding in this feature domain using coding based methods.
The authors claim that after experiments OrthoMark significantly improves the joint trade-off among
visual quality, robustness, and capacity.
However, I think that the description of encoding and decoding is rather unclear.

**Compliance With Llm Reviewing Policy:**

Affirmed.

**Key Questions For Authors:**

1. What is the advantage of having the orthogonal projection?

**Limitations:**

From Table 1, the approach is slightly better than known resutls for some categories.

**Strengths And Weaknesses:**

Soundness: The problem is interesting. But the results are a little incremental to known results.

---

> ### Author Rebuttal · Authors · 2026-03-31
>
> Thank you for the thoughtful review and for the Excellent rating on significance. We are glad that the core contribution resonated, and we address your questions below.
>
> ---
>
> **To provide a more comprehensive response, we have included additional results at the following anonymous link: https://anonymous.4open.science/r/Rebuttal-ICML-9170-BB112010/Rebuttal_ICML2026_9170.pdf**
>
> ---
>
> **Q1: What is the advantage of orthogonal projection?**
>
> The necessity of orthogonal projection arises from a fundamental challenge in multi-bit watermark embedding: when multiple bits are embedded simultaneously in a shared feature space, their carriers interact and distort one another’s decoding boundaries, a problem we call coding cross-talk. Orthogonal projection is the structural choice that eliminates this interference by design.
>
> Concretely, if two carrier vectors are not orthogonal, the quantization step for bit *i* shifts the projection of bit *j* onto its own carrier, distorting the decoding boundary for bit *j*. As the number of bits grows, these mutual distortions accumulate, and the entire decoding system collapses, which is exactly the behavior shown in Figure 1. Orthogonal carriers ensure that each bit’s projection is unaffected by the embedding of every other bit, making the per-bit channels independent.
>
> The experimental evidence is already presented in Table 6 of the paper. For convenience, we reproduce the relevant rows here:
>
> | Method | PSNR (dB) | AVG Accuracy (%) |
> |---|:---:|:---:|
> | Cor + QIM | 33.61 | 92.02 |
> | Oth + QIM (ours) | 39.26 | 99.59 |
>
> Switching from correlated to orthogonal carriers yields a 7.57-point gain in average accuracy and a 5.65 dB gain in PSNR simultaneously. The PSNR gain reflects a fundamental property: correlated carriers require larger perturbations to overcome cross-talk, degrading visual quality; orthogonal carriers preserve both robustness and image fidelity. This dual benefit directly validates the role of orthogonal projection.
>
> ---
>
> **Q2: The approach is slightly better than known results for some categories**
>
> We appreciate you raising this point; it prompted us to conduct experiments at higher message lengths to further substantiate our advantage in the high-capacity regime, which we present below.
>
> We would first like to note that the paper’s key contribution lies in the high-capacity regime: as Figure 1 shows, even without any distortion, all prior deep watermarking methods fail to decode reliably beyond a certain payload threshold, collapsing to near-chance accuracy, while OrthoMark maintains reliable decoding up to 32,768 bits. **This is not an incremental improvement over a working baseline; it is a qualitatively new capability that no prior deep watermarking method has achieved.**
>
> Additional experiment at 256-bit payload (Table 10, linked). Motivated by your insightful suggestion, we provide full robustness results at a 256-bit payload under the same 15 distortions used in Table 1, with all methods evaluated at PSNR ≈ 40 dB, to make our high-capacity robustness advantage directly concrete.
>
> At 256 bits, OrthoMark achieves 90.6% average accuracy, surpassing the strongest baseline (CIN, 84.6%) by 6.0 points. Notably, CIN, VideoSeal, and MBRS largely collapse on geometric distortions (C&R, Shear, Rotation, Elastic) at this payload, while OrthoMark maintains 75–88% accuracy across all geometric categories.
>
> Why the 64-bit gap appears modest. At a 64-bit payload, inter-bit interference is still limited, so strong baselines such as CIN and VideoSeal remain effective and the gap is naturally smaller. As payload grows, interference accumulates and the baselines degrade significantly, while OrthoMark maintains its advantage. The 64-bit result in Table 1 is best understood as a reference point; the primary contribution is most clearly demonstrated in the high-capacity regime.
>
> We will revise the paper’s framing to foreground the high-capacity comparison.

---

> > ### Author Rebuttal · Reviewer_ScsT · 2026-04-01
> >
> > My concerns have been adequately addressed.

---

> > > ### Author Response · Authors · 2026-04-04
> > >
> > > Thank you very much for carefully considering our rebuttal and for your follow-up acknowledgement. We sincerely appreciate your time and thoughtful evaluation. We are glad that our response was able to adequately address your concerns. Your comments and suggestions have been very helpful in improving the quality of our work.

---

### Official Review · Reviewer_suQ4 · 2026-03-13

**Soundness:** 3
**Presentation:** 2
**Significance:** 2
**Originality:** 3
**Overall Recommendation:** 3
**Confidence:** 4

**Summary:**

This paper identifies a key limitation in deep watermarking—decoding accuracy collapses as payload increases due to "coding cross-talk" from non-orthogonal bit carriers and encoder-decoder misalignment. To address this, the authors propose OrthoMark, a hybrid framework that combines a deep robust feature extractor (based on invertible networks in the wavelet domain) with a structured coding module (orthogonal projection + QIM). This design decouples representation learning from message encoding, enabling high capacity without sacrificing robustness or visual quality. Experiments show OrthoMark achieves near-perfect accuracy at up to 32,768 bits and superior robustness across diverse distortions, outperforming existing SOTA methods.

**Compliance With Llm Reviewing Policy:**

Affirmed.

**Key Questions For Authors:**

See weakness

**Limitations:**

No. The authors primarily focus on the advantages of their method and do not adequately discuss its limitations. I suggest adding a "Limitations" section in the final version to transparently address key concerns, particularly the significant inference-stage computational cost: as noted in the questions above, the iterative optimization process ($T=3000$) is substantially more expensive than single forward-pass methods, posing a major limitation for real-time or resource-constrained applications.

**Strengths And Weaknesses:**

Strengths:
1. In general, this paper identifies a critical but overlooked issue in deep watermarking—high-capacity performance collapse—and provides a systematic analysis attributing it to the "curse of dimensionality" and "coding cross-talk."
2. The introduction of the Signal-to-Interference Ratio (SIR) provides a rigorous theoretical foundation that explains both the failure of existing methods and the design rationale of OrthoMark.
3. Comprehensive experiments show OrthoMark achieves state-of-the-art performance, particularly in high-capacity scenarios (up to 32,768 bits) and across a diverse set of 15 distortions, with extensive ablation studies validating each component.

Weaknesses:
1. Significant Inference Overhead: The inference process requires $T=3000$ iterative optimization steps, which is computationally prohibitive for practical deployment, especially on high-resolution images or video. This stands in stark contrast to most deep watermarking methods that operate with a single forward pass, and the paper lacks discussion on this fundamental trade-off between quality and speed.
2. Insufficient Validation for Video Watermarking: Despite claiming video adaptability via "temporal watermark pooling," the paper lacks quantitative evaluations on standard video benchmarks under typical video distortions (e.g., H.264 compression, temporal attacks, frame dropping). The image-based experiments alone are insufficient to demonstrate its effectiveness in the video domain.

---

> ### Author Rebuttal · Authors · 2026-03-31
>
> We sincerely thank you for the incisive summary and constructive feedback.
>
> **Additional results: https://anonymous.4open.science/r/Rebuttal-ICML-9170-BB112010/Rebuttal_ICML2026_9170.pdf**
>
> ---
>
> **W1:** We acknowledge the trade-off: delivering reliable decoding at 32,768 bits requires iterative optimization, and this encoding cost is real. However, as you mentioned in your summary, this paper addresses "a critical but overlooked issue." We further discussed the efficiency concern from four angles:
>
> **(1) Significance of the contribution.** Before OrthoMark, no amount of brute-force scaling could reach this capacity: ChunkySeal scales VideoSeal's embedder by 90× and extractor by 23× yet only achieves a 4× payload gain (256→1024 bits). We are the **first to establish a viable trade-off between encoding time and high capacity, strong robustness, and visual quality.** By identifying the theoretical root cause (SIR analysis) and providing a principled solution, we bridge a gap that parameter scaling alone could not close, laying both the theoretical foundation and a working practical framework for future work to build upon.
>
> **(2) Practical with smaller T.** T=3,000 reaches diminishing returns (Table 7); the practical setting is T=600, which already surpasses every baseline in Table 1. Full latency measurements (NVIDIA A100, 128×128 RGB) are in **Table 8 (linked).** On the encoding side, we acknowledge higher latency than single-pass methods. However, among optimization-based methods, OrthoMark is the most efficient: **2.9× faster than SSL and 6.2× faster than LISO**, with strictly stronger robustness. On the decoding side, our decoder runs in a single forward pass (roughly 0.9 ms/image), **comparable to FIN and MBRS.**
>
> **(3) Application scenarios.** Important real-world workflows exist where high-capacity robustness is the primary requirement and the encoding cost is acceptable.
>
>   - **Media provenance.** A news agency certifying photographs for long-term redistribution requires sufficient payload for structured provenance metadata. Each image is signed once but verified many times. Encoding is a one-time offline process where latency is not time-critical (≈1s); the bottleneck lies in high-throughput verification, where our 0.9 ms decoding is well suited.
>
>   - **Generative AI governance.** Embedding rich forensic metadata for content attribution demands both large capacity and strong robustness. No prior method jointly satisfies this; OrthoMark is the first. Our ≈1s encoding does not introduce a significant bottleneck relative to the latency of generative models themselves.
>
> **(4) Future direction.** Amortizing the iterative optimization into a single-pass encoder via knowledge distillation is the natural next step and the primary future direction of this work.
>
> ---
>
> **W2:** Our paper focuses entirely on image watermarking. Video watermarking is a distinct research problem involving temporal consistency, inter-frame compression (H.264/HEVC), and motion-based distortions, which require fundamentally different treatment. While the OrthoMark framework may eventually inform video watermarking research, this problem is **beyond the scope of the present submission**. We will add a scope statement in the introduction to make this clear.
>
> ---
>
> **Limitation:** Thank you for this valuable suggestion. We will foreground the latency trade-off in the Abstract and Introduction, and add a dedicated Limitations section discussing the inference cost.

---

> > ### Author Rebuttal · Reviewer_suQ4 · 2026-04-06
> >
> > The author has addressed my concerns through detailed explanations and experimental results to a certain extent.

---

> > > ### Author Response · Authors · 2026-04-07
> > >
> > > We sincerely thank you for selecting "(a) Fully resolved" and are honored that your concerns have been adequately addressed. Your constructive guidance has substantially improved our manuscript. **However, we notice that your score remains at Weak Reject.** We wonder if there are remaining concerns that lead to this decision? We would be glad to address them during the discussion period and welcome any additional feedback.

---

### Decision · Program_Chairs · 2026-04-30

**Decision:**

Accept (regular)

**Comment:**

In this paper, the authors propose OrthoMark, a framework that decouples robust feature extraction from message encoding, to leverage the strengths of deep learning–based watermarking and quantization index modulation (QIM).
According to the reviewers' concerns and authors' responses, all the reviewers basically appreciate the contributions of this work, including (1) OrthoMark achieves state-of-the-art performance; (2) The theoretical analysis is executed exceptionally well; (3) Embedding orthogonal projection and QIM into a deep latent space is a clever reconstruction of the existing paradigm; (4) The contribution is significant; and (5) The concerns from all reviewers have been adequately addressed.
Based on the above, the AC tends to recommend the acceptance of this paper!